# Discovery and rational engineering of PET hydrolase with both mesophilic and thermophilic PET hydrolase properties

Hwaseok Hong [1,4], Dongwoo Ki [1,4], Hogyun Seo [1], Jiyoung Park [1], Jaewon Jang [2] & Kyung-Jin Kim [1,3] ✉

Excessive polyethylene terephthalate (PET) waste causes a variety of problems. Extensive research focused on the development of superior PET hydrolases for PET biorecycling has been conducted. However, template enzymes employed in enzyme engineering mainly focused on *Is*PETase and leaf-branch compost cutinase, which exhibit mesophilic and thermophilic hydrolytic properties, respectively. Herein, we report a PET hydrolase from *Cryptosporangium aurantiacum* (*Ca*PETase) that exhibits high thermostability and remarkable PET degradation activity at ambient temperatures. We uncover the crystal structure of *Ca*PETase, which displays a distinct backbone conformation at the active site and residues forming the substrate binding cleft, compared with other PET hydrolases. We further develop a *Ca*PETase[M9] variant that exhibits robust thermostability with a $T_m$ of 83.2 °C and 41.7-fold enhanced PET hydrolytic activity at 60 °C compared with *Ca*PETase[WT]. *Ca*PETase[M9] almost completely decompose both transparent and colored post-consumer PET powder at 55 °C within half a day in a pH-stat bioreactor.

Plastics are synthetic polymers, with advantages of chemical resistance, light weight, and low cost. They have been extensively used worldwide since the 1950s and have become essential in our daily lives[1,2]. However, because plastics do not naturally decompose, billions of tons of plastic waste in landfills, floating in waste islands in the ocean, and dispersed as microplastics have led to severe global pollution[3–7]. Governments and plastic manufacturers around the world are aware of these issues, and research and development focused on chemical and biological recycling strategies for plastic waste have accelerated in recent years[8–10].

Polyethylene terephthalate (PET), a polyester composed of units of terephthalic acid (TPA) and ethylene glycol, is a widely used plastic packaging material and is relatively easy to recycle. Studies on the biological degradation of PET have been actively conducted in the last two decades[8,11–13]. PET biorecycling technology includes hydrolysis of PET by microorganisms or enzymes and its conversion into high-value-added chemicals, which ultimately establishes a circular economy for PET[11,14–16].

To date, various enzymes capable of hydrolyzing PET have been discovered and characterized biochemically and structurally. In particular, PETase from *Ideonella sakaiensis* 201-F6 (*Is*PETase) exhibits the highest PET hydrolytic activity at ambient temperatures and is considered a promising enzyme for PET waste treatment[17]. Accordingly, various enzyme engineering studies are currently underway to improve the PET hydrolysis efficiency and thermal stability of *Is*PETase, and variants with improved performance have been reported[18–24]. Recently, metagenome-derived leaf-branch compost cutinase (LCC), which exhibits thermophilic properties, has been discovered[25]. Its variant has shown to exhibit high PET depolymerization activity in a bioreactor system at 72 °C, rendering it a candidate for biological PET recycling[26]. Moreover, efforts for discovering new PET-degrading enzymes are being made, and enzymes from various microorganisms

[1]School of Life Sciences, BK21 FOUR KNU Creative BioResearch Group, KNU Institute for Microorganisms, Kyungpook National University, Daegu 41566, Republic of Korea. [2]Institute of Biotechnology, CJ CheilJedang Co., Suwon-si, Gyeonggi-do 16495, Republic of Korea. [3]Zyen Co, Daegu 41566, Republic of Korea. [4]These authors contributed equally: Hwaseok Hong, Dongwoo Ki. ✉e-mail: kkim@knu.ac.kr

and environmental metagenomes, such as *Thermobifida fusca* cutinases (*Tf*Cut1,2)[27], *Thermomonospora curvata* DSM43183 cutinases (*Tcur*1278,0390)[28], Bacterium HR29 (BhrPETase)[29], *Fusarium solani pisi* (*Fs*Cut)[30], *Humicola insolens* (HiC)[31,32], PET2[33], PET5[33], PET6[33], and PHL7[34] have been reported. And recently, thermotolerant PET hydrolases were discovered by genome mining in bioinformatics[35].

For feasible application of enzymatic PET hydrolysis, the inherently high catalytic activity of PET hydrolases is crucial factor as a starting point. Also, exploiting the thermophilic PET hydrolase properties is also an efficient strategy for development of superior PET hydrolases, because high-temperature operation near the glass transition temperature of PET material is favorable to the PET degradation performance[36]. Thus, the discovery of promising PET-degrading enzymes that have both high catalytic activity and high thermostability properties is strongly needed to provide insight into the catalysis reaction and expand the scope of efficient PET biocatalysts.

There is a rapid progress in research on developing efficient PET-degrading enzymes, and numerous variants have emerged. Nonetheless, recently developed PET hydrolases do not outperform the previously reported enzymes in terms of enzymatic catalysis and thermostability. Thus, template enzymes used for enzyme engineering are mainly focused on *Is*PETase and LCC, which have mesophilic and thermophilic hydrolytic properties, respectively. In the present study, we report a PET hydrolase from *Cryptosporangium aurantiacum* (*Ca*PETase) that exhibits remarkable PET degradation activity at ambient temperatures and high thermostability. The crystal structure of *Ca*PETase indicates that the enzyme has a unique active site compared with other known PET hydrolases. Through structure-guided rational protein engineering, we developed a robust *Ca*PETase variant that exhibits markedly enhanced enzyme activity and thermostability. The applicability of this variant in the recycling industry was further demonstrated by evaluating its activity with a pH-stat bioreactor.

## Results

### Discovery of a *Ca*PETase with high PET hydrolytic activity and thermostability

To discover a PET hydrolase, we performed a sequence homology analysis using the National Center for Biotechnology Information (NCBI) database and selected 10 PETase candidates (see "Methods" for details). A phylogenetic tree was constructed for the 10 selected PETase candidates and 17 reported PET hydrolases. The 27 enzymes were divided into two groups: one group contained mesophilic enzymes, such as *Is*PETase, and the other group contained thermophilic enzymes, such as *Tf*Cut2 and LCC (Fig. 1a, Supplementary Fig. 1 and Supplementary Table 1). The phylogenetic tree was further separated into 10 subgroups, and 3 selected PETase candidates, namely, RZL00883.1, KOX11336.1, and SHM40309.1, formed discrete lineages with low phylogenetic relationships to the reported PET hydrolases (Fig. 1a). To characterize the 10 selected PETase candidates (PCs), we first attempted to produce them in a signal peptide-truncated form. Eight of the 10 PETase candidates were successfully produced, except for KOX11336.1 and MAM88718.1. We also measured the melting temperature ($T_m$) of the eight PETase candidates to determine the thermostability of these enzymes. The candidates exhibited a range of $T_m$ values from 38.6 °C to 70.5 °C (Fig. 1b and Supplementary Fig. 2). We then checked PET hydrolytic activity of the eight PETase candidates at a wide range of temperatures from 30 to 60 °C using several PET samples, such as post-consumer transparent PET powder (PC-PET$^{Transparent}$), semi-crystalline PET powder (Cry-PET, Goodfellow Cambridge Ltd) (Cat. No. ES306000), and amorphous PET film (AF-PET, Goodfellow Cambridge Ltd) (Cat. No. ES301445). Among these candidates, PC3, PC7, PC8 and PC10 showed relatively low or undetectable levels of PET hydrolytic activity across the tested conditions (Fig. 1c and Supplementary Fig. 3). PC6, which have the highest $T_m$ value of 70.5 °C, produced only negligible amounts of PET hydrolysis products

at 30 °C, but showed the optimal PET hydrolytic activity at 50 °C (Fig. 1b, c and Supplementary Figs. 2, 3). PC4 and PC5 showed relatively high PET hydrolytic activity at 50 °C and 40 °C, respectively (Fig. 1c and Supplementary Fig. 3). Surprisingly, compared to other PETase candidates, PC2 exhibited significantly high PET hydrolytic activity across a broad range of reaction conditions. Notably, PC2 exhibited superior activity at 30 °C and produced the highest amount of PET hydrolysis products across all three PET substrates conditions compared to other candidates. (Fig. 1c and Supplementary Fig. 3). In addition, the pH profile results for the eight PETase candidates also showed that PC2 showed the highest level of PET hydrolytic activity (Supplementary Fig. 4). It is noteworthy that PC2 showed remarkable PET hydrolytic activity compared with the other enzymes, and also exhibited high thermostability with a $T_m$ value of 66.8 °C (Fig. 1b, c and Supplementary Fig. 2). Moreover, PC2 had the highest soluble expression level compared with the other enzymes (Fig. 1b). These results indicate that PC2 has excellent properties for efficient PET degradation, including enzyme activity, thermostability, and protein expression levels. Thus, we selected PC2 (accession code: SHM40309.1, PETase from *Cryptosporangium aurantiacum*, *Ca*PETase) as the most robust PET hydrolase among the eight PETase candidates tested. The measurements of changes of the $T_m$ value and activity by addition of metal ions showed that PC2 is not a metal ion-dependent enzyme (Supplementary Fig. 5).

Next, we compared the PET hydrolytic activity of *Ca*PETase with that of well-known PET hydrolases, such as *Is*PETase, *Tf*Cut2, and LCC, over a broad temperature range from 30 to 60 °C using Cry-PET as a substrate. In reactions at 30 °C, *Ca*PETase showed significantly higher PET hydrolytic activity than LCC and *Tf*Cut2 (Fig. 1d and Supplementary Fig. 3). Moreover, *Ca*PETase exhibited 1.4-fold higher activity than *Is*PETase, which is known to have the highest PET hydrolytic activity at ambient temperature among the reported PET hydrolases (Fig. 1d)[17]. In particular, the PET hydrolytic activity of *Ca*PETase was 3.1-fold higher than that of *Is*PETase at 40 °C (Fig. 1d), likely because *Ca*PETase has much higher thermostability than *Is*PETase. However, the PET hydrolytic activity of *Ca*PETase at 60 °C dramatically decreased and reversed compared with that of LCC at temperatures of 50 °C and 60 °C (Fig. 1d). These results indicate that *Ca*PETase is a promising PET hydrolase that exhibits high PET decomposition ability and thermostability. Considering that it is important to make improvements without the loss of enzyme activity and thermostability in the development of superior PET-degrading enzymes[37], we propose that *Ca*PETase represents a more efficient template enzyme for enzyme engineering than other enzymes with extreme mesophilic and thermophilic properties, such as *Is*PETase and LCC, respectively.

### Unique structural features of *Ca*PETase

To provide a structural basis for high PET hydrolytic activity of *Ca*PETase, we determined its crystal structure at a resolution of 1.36 Å (Supplementary Table 2). *Ca*PETase shows an α/β hydrolase fold and a nine-stranded β-sheet at the center surrounded by six α-helices and two $3_{10}$-helices (Fig. 2a and Supplementary Fig. 6). Sequence-independent pairwise superposition of *Ca*PETase with three distinctive PET hydrolases, namely, *Is*PETase, LCC, and *Tf*Cut2, generated global root mean square deviation values of 0.69, 0.65, and 0.53 Å, respectively. Formation of one conserved disulfide bond (DS, C279/C297) and lack of an extended loop in the substrate binding site of *Ca*PETase suggest that the enzyme originated from an ancestor of *Tf*Cut2 and LCC rather than *Is*PETase (Supplementary Fig. 7). Interestingly, *Ca*PETase exhibits a somewhat different backbone structure at the active site compared with other PET hydrolases (Supplementary Fig. 8). Because the structural comparison using a Cartesian coordinate system is known to be subjective for distinguishing the detailed structural differences of the main chains[38], we further analyzed the φ-ψ torsion angles of the main chains of these four PETases

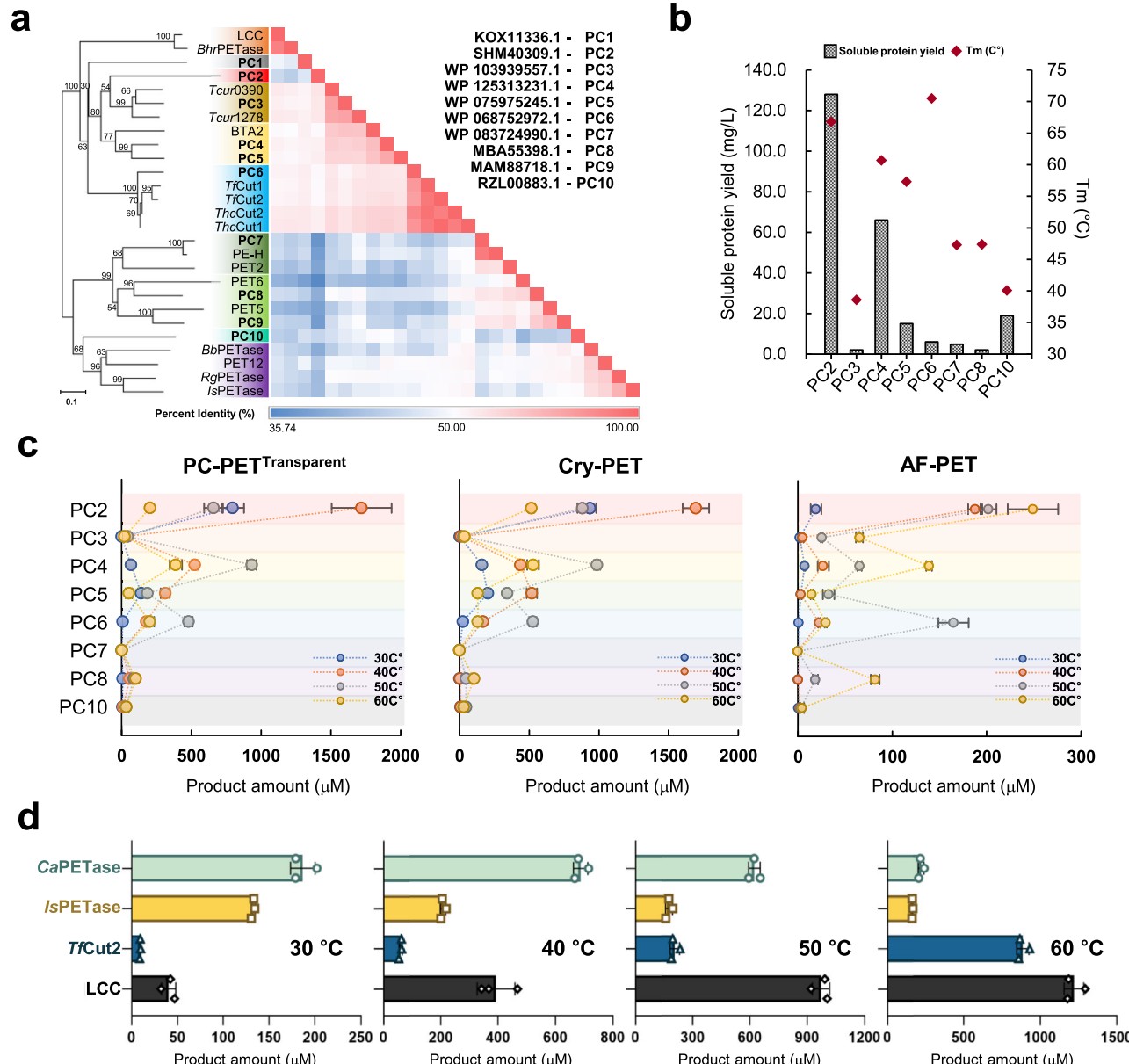

**Fig. 1 | The discovery a PET hydrolase and initial characterizations of *Ca*PETase.** **a** Maximum likelihood phylogenetic tree and percentage identity matrix of the 10 selected PETase candidate (PC1–PC10) and 17 reported PET hydrolase sequences. Bootstrap values for 1000 replications are shown at the branching edges. The colored bar represents the level of the percent identity of the enzymes, and detailed percent identity values are listed in Supplementary Fig. 31. **b** Protein yield and the $T_m$ values of the 8 PCs. **c** PET hydrolytic activity of the eight PCs. The reaction was performed with post-consumer transparent PET powder (PC-PET$^{Transparent}$, 15 mg mL$^{-1}$ with 500 nM enzyme), semi-crystalline PET powder (Cry-PET, 15 mg mL$^{-1}$ with 2 μM enzyme), and amorphous PET film (AF-PET, 15 mg mL$^{-1}$ with 2 μM enzyme) in 50 mM Glycine-NaOH pH 9.0 buffer at various temperatures (30 °C, 40 °C, 50 °C, 60 °C) for 3 days. Reactions were performed in triplicate; Data are presented as mean values ± SD. **d** Comparison of the PET hydrolytic activity of *Ca*PETase, *Is*PETase, LCC, and *Tf*Cut2. The reaction was conducted with Cry-PET (15 mg mL$^{-1}$ with 2 μM enzyme) in 50 mM Glycine-NaOH (pH 9.0) at various temperatures (30 °C, 40 °C, 50 °C, 60 °C) for 12 h. Reactions were performed in triplicate; Data are presented as mean values ± SD.

(Supplementary Fig. 9) and found that there were local differences in backbone torsion angles between *Ca*PETase and other PET hydrolases (Fig. 2a and Supplementary Fig. 10). Interestingly, *Ca*PETase also showed significant differences in the backbone torsion angles at the five connecting loops (β3–α1, β4–α2, β6–β7, β7–α4, and β8–α5) that form an active site, whereas comparisons of the corresponding loops between the other three PET hydrolases exhibited less differences (Fig. 2a and Supplementary Fig. 11), suggesting that *Ca*PETase has a unique active site conformation. There were some differences in the network of residues extending from the active site to the nearby spatial environment compared with that of the other PET hydrolases. Among them, we observed unique differences affecting the backbone torsion

angles of these loops. Near the β3–α1 loop, distinct residues positioned in the β4–α2 loop and a W105–L108–G124 network force, which form a unique side-chain internal network, appear to influence the conformation of the β3–α1 loop (Supplementary Fig. 12). In fact, the β3–α1 loop has high root mean square fluctuation values near the active sites of other PET hydrolases in molecular dynamic simulations[39,40]. A unique A192–G212–F248 network is formed under the β8–α5 loop, where catalytic H246 is located (Supplementary Fig. 13). At the corresponding F248 position in *Ca*PETase, LCC and *Tf*Cut2 have an alanine residue, whereas *Is*PETase has a cysteine residue that forms a second disulfide bond. Therefore, the positioning of a bulky F248 might cause significant torsional differences in the β8–α5

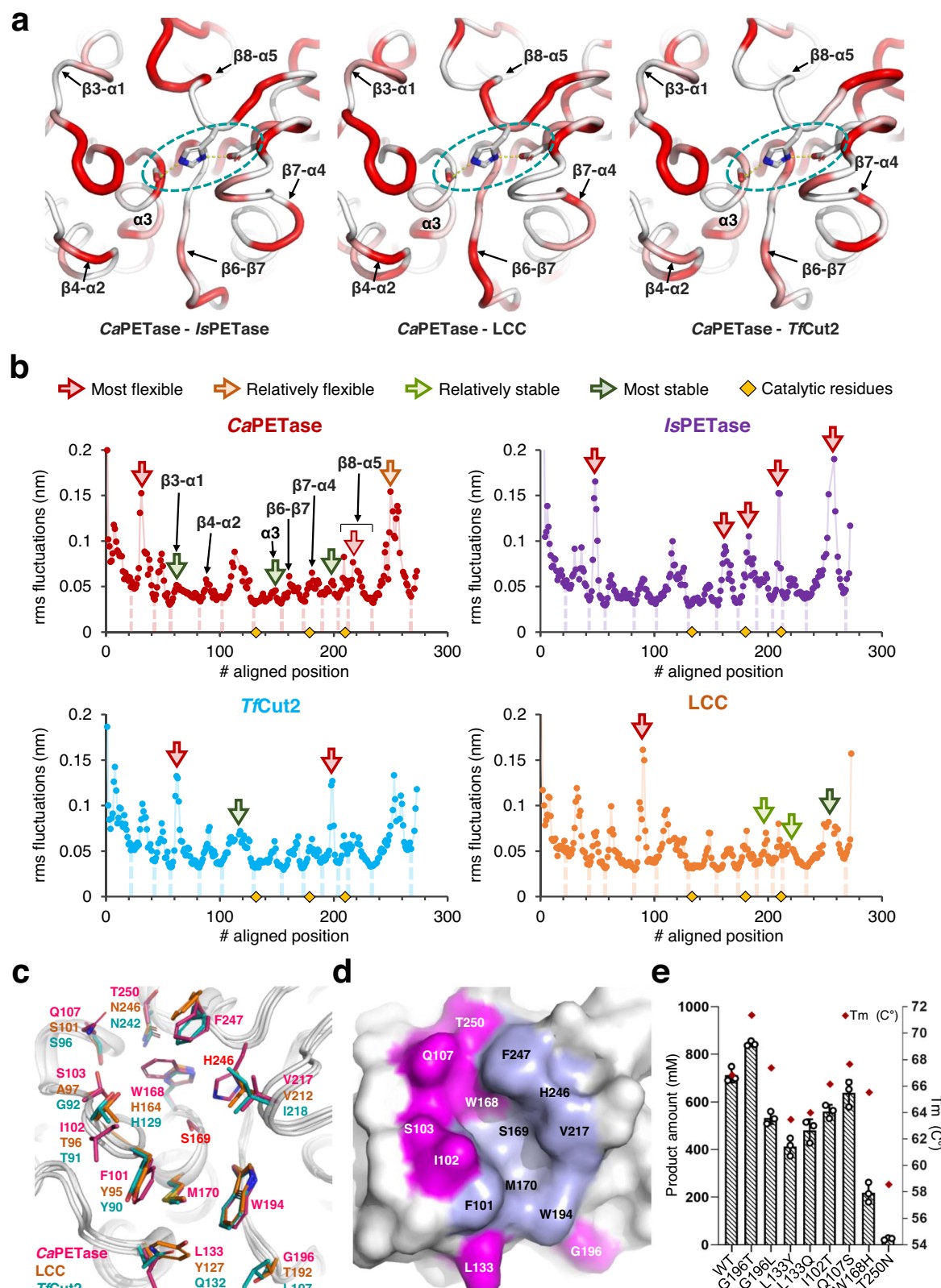

loop and β7–α4 loop of *Ca*PETase (Supplementary Fig. 13). Finally, an R176–W200–F209 network appears to trigger conformational differences in the α3-helix and β6–β7 loop (Supplementary Fig. 14). Importantly, the α3-helix contains the catalytic S169, and the β6–β7 loop was previously annotated as a wobbling tryptophan-containing loop in PETase from *Rhizobacter gummiphilus* (Supplementary Fig. 14)[41]. We further analyzed backbone fluctuations of these four PET hydrolases using molecular dynamic simulations and *Ca*PETase exhibits quite unique backbone fluctuation profile (Fig. 2b and Supplementary Fig. 15). *Ca*PETase has more stable β6–β7 and β7–α4 loops than mesophilic *Is*PETase, and particularly, the enzyme shows high stability at the front region of the β8–α5 catalytic loop where H246 is located (Fig. 2b). However, the end region of β8–α5 which corresponds to the extended loop of *Is*PETase, and the front region of β3–α1 loop showed

**Fig. 2 | Unique structural features of *Ca*PETase. a** Comparison of the backbone torsion angle differences between *Ca*PETase and *Is*PETase, LCC, and *Tf*Cut2. The structure of five connecting loops forming the active site of *Ca*PETase is displayed as a putty tube representation of the same diameter in PyMoL. The structure is colored according to the Euclidean distance values between the two Ramachandran points of the aligned residues. Colors of white to red designate low to high Euclidean distance values, respectively. The catalytic triad of *Ca*PETase is shown as a stick model with a cyan-color circle. **b** MD simulations show unique backbone fluctuation profile of *Ca*PETase. Cα atom root-mean-square fluctuations (RMSF, Å) of the *Ca*PETase, *Is*PETase, *Tf*Cut2, LCC during MD simulations. **c** Comparison of the residues forming the substrate binding cleft of *Ca*PETase, LCC, and *Tf*Cut2. The highlighted residues are shown as a stick model. **d** Distinct residues in the substrate binding site of *Ca*PETase. Distinct and conserved residues are presented in magenta and light blue, respectively. **e** PET hydrolytic activity of the variants. PC-PET$^{Transparent}$ (15 mg mL$^{-1}$) were incubated with 500 nM enzymes at 40 °C for 24 h in 50 mM Glycine-NaOH buffer pH 9.0. Total amount of released products and the $T_m$ value of the variants are shown as bars and red-colored dots, respectively. Reactions were performed in triplicate; Data are presented as mean values ± SD.

the highest and lowest flexibility among the four homologs, respectively (Fig. 2b). To our interest, the differences of the backbone fluctuation profile was localized exactly to the unique internal network affecting the backbone torsion angles of these loops. Thus, we believe that the unique backbone conformation of *Ca*PETase allows the enzyme to maintain high activity while stabilizing several flexible loops of the mesophilic PET hydrolase.

In addition to the unique backbone conformation at the active site, residues forming the substrate binding cleft of *Ca*PETase showed significant differences compared with other thermophilic PET hydrolases (Fig. 2c, d). In the vicinity of the wobbling W194, *Ca*PETase possesses unique G196 and L133 residues, where highly conserved residues are located in other PET hydrolases (Fig. 2c, d and Supplementary Fig. 7). Mutating these residues to the corresponding residues in other PET hydrolases, such as G196L, L133Y, and L133Q, had a negative effect on enzyme activity and/or thermostability (Fig. 2e). However, the G196T mutation exhibited enhanced thermostability (Fig. 2e), which may result from the formation of a hydrogen bond between G196T and N195. *Ca*PETase also contains a unique I102 residue in the β3−α1 loop showing the largest torsion differences, whereas other PET hydrolases contain a highly conserved threonine residue at the corresponding position, which probably enables *Ca*PETase to form a relatively wider substrate binding cleft (Fig. 2c, d and Supplementary Fig. 16). Replacement of I102 with threonine resulted in decreased enzyme activity, confirming that the residue contributes to high enzyme activity (Fig. 2e). Furthermore, *Ca*PETase has unique residues, such as Q107, W168, and T250, at the regions of the β3−α1 loop, β8−α5 loop, and α3, whereas most of the corresponding residues are highly conserved in other thermophilic PET hydrolases (Fig. 2c, d and Supplementary Fig. 7). Mutating these residues to the conserved residues in other thermophilic PET hydrolases decreased enzymatic activity and/or stability, indicating that the combined positioning of these residues is necessary to create an optimal substrate binding site for *Ca*PETase with a unique shape and polarity (Fig. 2e). One exception was the Q107S mutation, which resulted in no noticeable differences in enzyme activity or thermostability (Fig. 2e). Taken together, we suggest that along with unique backbone torsion angles, the positioning of distinct residues at the substrate binding site enable *Ca*PETase to form an optimal substrate binding site for high PET hydrolytic activity.

## Rational protein engineering of *Ca*PETase

Although *Ca*PETase has high PET hydrolytic activity and thermostability, its performance is still insufficient for industrial applications. We conducted rational protein engineering of *Ca*PETase to further enhance the PET hydrolytic activity and thermostability of the enzyme using various strategies, such as introducing disulfide bonds and hydrogen bonds and modifying the protein surface charge (Supplementary Fig. 17). The thermostability of the variants was monitored by measuring $T_m$ values, and the PET hydrolytic activity of the variants was measured using post-consumer transparent PET powder (PC-PET$^{Transparent}$) at ambient temperature (40 °C). We introduced four disulfide bonds, namely, G76C/A143C (DS1), L180C/A202C (DS2), T204C/R233C (DS3), and R242C/S291C (DS4). The DS2 and DS4 mutations increased the $T_m$ value by approximately 3 °C, whereas the DS1 and DS3 mutations decreased the $T_m$ value compared with *Ca*PETase$^{WT}$

(Fig. 3a and Supplementary Fig. 18). Moreover, the introduction of the DS2 and DS4 mutations increased PET hydrolytic activity by more than 20% compared with *Ca*PETase$^{WT}$ (Fig. 3a and Supplementary Fig. 18). These results indicate that the DS2 and DS4 mutations were successfully formed in *Ca*PETase$^{WT}$ and exerted positive effects on enzyme activity and thermostability. We also attempted to improve the thermostability of *Ca*PETase by introducing noncovalent bonds, such as hydrogen bonds and salt bridges, and designed seven mutations, namely, V129T (NC1), P136S (NC2), A192T (NC3), R198K (NC4), V203T (NC5), A252N (NC6), and A257S(NC7). Of these, the NC1 and NC4 mutations increased $T_m$ values by approximately 2 °C and enhanced PET hydrolytic activity by 30% compared with *Ca*PETase$^{WT}$ (Fig. 3a and Supplementary Fig. 18). Finally, in an attempt to improve the protein adsorption ability to the PET surface by modifying the protein surface charge, we designed five mutations to render the protein surface hydrophobic, i.e., N109A (HP1), R151A (HP2), R157A (HP3), R160A (HP4), and R233A (HP5), and four mutations to render the protein surface positive, i.e., T86R (SC1), A155R (SC2), T275R (SC3), and M294R (SC4). Unfortunately, most mutations did not show significant changes or even negative effects on thermostability or enzyme activity; however, the HP1 mutation increased the $T_m$ value by 3.2 °C, and the SC2 mutation enhanced PET hydrolytic activity by 20% compared with *Ca*PETase$^{WT}$ (Fig. 3a and Supplementary Fig. 18). Taken together, we introduced eight point-mutations that resulted in improved thermostability and PET hydrolytic activity, i.e., DS2, DS4, NC1, NC4, HP1, and SC2, among the 20 rationally designed mutations tested (Fig. 3a and Supplementary Fig. 18). There were also ambiguous mutations that only improved enzyme activity or thermostability, such as NC2, HP2, and HP5. We excluded these mutations from the final selection to develop a much superior variant without compromising enzymatic activity or thermostability (Fig. 3a and Supplementary Fig. 18)[37].

## Combinatorial mutations of *Ca*PETase

We sequentially integrated the six mutations described above to develop a superior *Ca*PETase variant with higher thermostability and PET hydrolytic activity. First, we combined the DS2 and DS4 mutations, and the resulting *Ca*PETase$^{DS2/DS4}$ variant showed a synergistic effect on thermostability with a $T_m$ value of 74.3 °C ($\Delta T_m = 7.4$ °C) (Fig. 3b and Supplementary Fig. 19). Moreover, the variant enhanced PET hydrolytic activity by 1.35- and 4.45-fold at 40 °C and 60 °C, respectively, compared with *Ca*PETase$^{WT}$ (Fig. 3b and Supplementary Fig. 19). Next, we set up *Ca*PETase$^{DS2/DS4}$ as a scaffold for the next combination. We integrated the NC1/NC4, HP1, and SC2 mutations individually into *Ca*PETase$^{DS2/DS4}$ using our engineering strategy. The addition of the NC1/NC4 mutation increased the $T_m$ value significantly by 3.9 °C and increased PET hydrolytic activity at both 40 °C and 60 °C (Fig. 3b and Supplementary Fig. 19). It showed 3.8- and 17-fold enhanced activity at 60 °C compared with *Ca*PETase$^{DS2/DS4}$ and *Ca*PETase$^{WT}$, respectively (Fig. 3b and Supplementary Fig. 19). The addition of the HP1 mutation increased the $T_m$ value by 2.7 °C and enhanced PET hydrolytic activity by 1.7-fold at 60 °C compared with *Ca*PETase$^{DS2/DS4}$ (Fig. 3b and Supplementary Fig. 19). When the SC2 mutation was integrated into the *Ca*PETase$^{DS2/DS4}$ variant, we observed no noticeable improvements in activity and thermostability; however, there was a slight increase in PET hydrolytic activity at 60 °C (Fig. 3b and Supplementary Fig. 19).

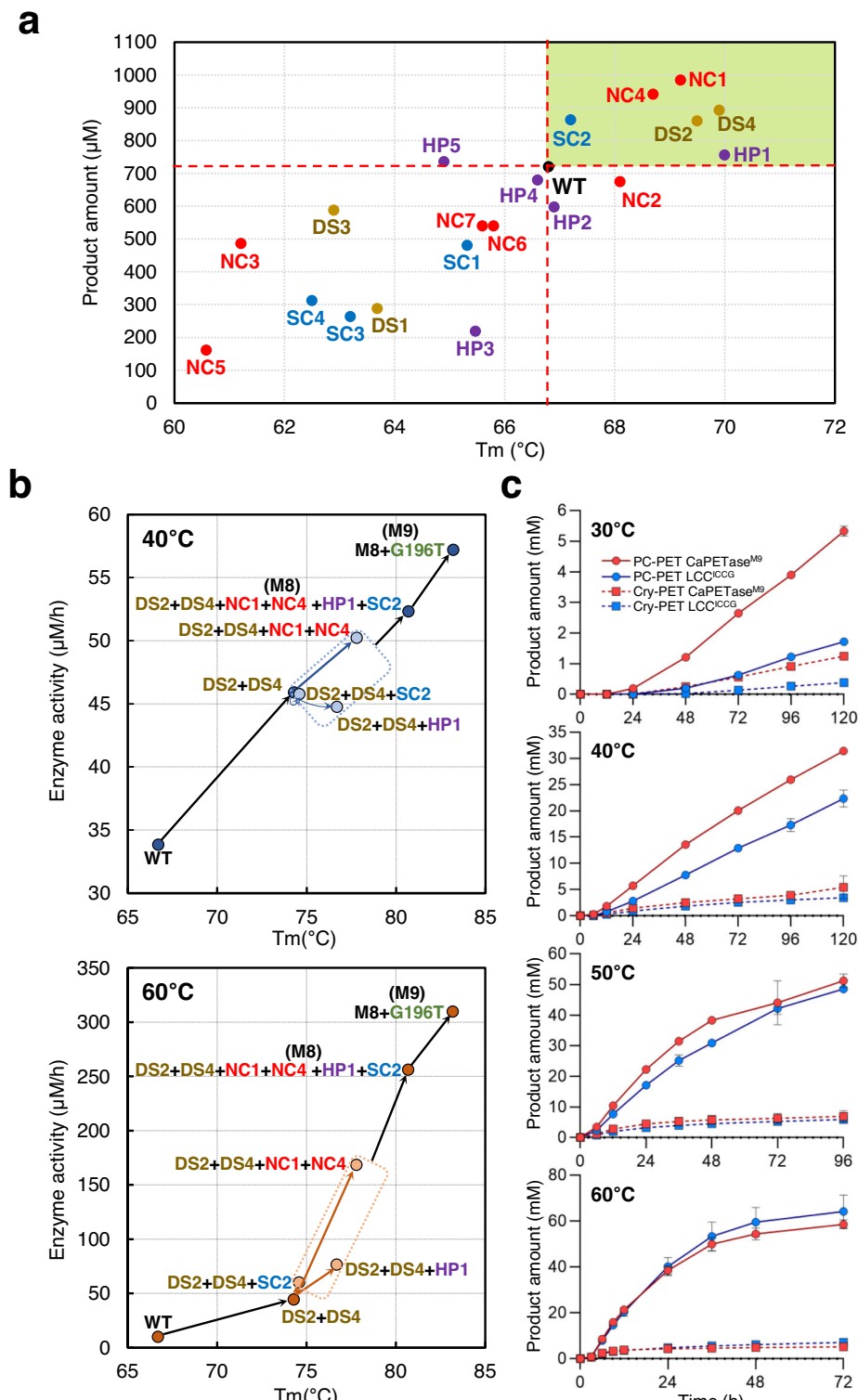

**Fig. 3 | Rational engineering of *Ca*PETase. a** Single-point mutations of *Ca*PETase. Released PET hydrolysis products and the $T_m$ values of the single-point mutations are presented. The reactions were performed using 500 nM enzymes with post-consumer transparent PET powder (PC-PET$^{Transparent}$, 15 mg mL$^{-1}$) as the substrate in 50 mM Glycine-NaOH buffer (pH 9.0) for 24 h at 40 °C. Reaction was carried out in triplicate; error bars represent the s.d. of the replicate measurement. **b** Combinatorial mutations of *Ca*PETase. PET hydrolytic activity of the variants generated using the combinatorial strategy. Released PET hydrolysis products per hour and the $T_m$ values of the combinatorial variants are presented. The reactions

were performed using 500 nM enzymes with PC-PET (15 mg mL$^{-1}$) as the substrate in 100 mM Glycine-NaOH buffer (pH 9.0) at 40 °C for 24 h and 60 °C for 6 h, respectively. Reaction was carried out in triplicate; error bars represent the s.d. of the replicate measurement. **c** Comparison of PET hydrolysis activity between *Ca*PETase$^{M9}$ and LCC$^{ICCG}$ at various temperatures. The reactions were carried out at different temperatures using PC-PET$^{Transparent}$ (12.5 mg mL$^{-1}$) with 1 μM enzyme and Cry-PET (12.5 mg mL$^{-1}$) with 4 μM enzyme under the 200 mM Glycine-NaOH buffer pH 9.0. Reactions were performed in triplicate; Data are presented as mean values ± SD.

These results suggest that all four mutations (NC1, NC4, HP1, and SC2) had a positive effect on the thermostability and activity of $Ca$PETase[DS2/DS4]; thus, we combined the four mutations into $Ca$PETase[DS2/DS4] to generate $Ca$PETase[DS2/DS4/NC1/NC4/HP1/SC2] ($Ca$PETase[M8]). Surprisingly, when all four mutations were added to $Ca$PETase[DS2/DS4], a synergistic effect on thermostability and enzyme activity was observed, and $Ca$PETase[M8] exhibited significantly enhanced thermostability with a $T_m$ value of 80.7 °C and 1.5- and 25.8-fold enhanced PET hydrolytic activity at 40 °C and 60 °C, respectively, compared with $Ca$PETase[WT] (Fig. 3b and Supplementary Fig. 19).

As mentioned above, the G196T mutation resulted in positive effects on both thermostability and enzyme activity (Fig. 2e); thus, we finally generated $Ca$PETase[DS2/DS4/NC1/NC4/HP1/SC2/G196T] ($Ca$PETase[M9]) by integrating the G196T mutation into $Ca$PETase[M8]. $Ca$PETase[M9] exhibited a $T_m$ value of 83.2 °C, which corresponds to a 16.7 °C increase in $T_m$ compared with $Ca$PETase[WT]. Moreover, the PET hydrolytic activity of $Ca$PETase[M9] increased by 1.7- and 31.2-fold at 40 °C and 60 °C, respectively, compared with $Ca$PETase[WT] (Fig. 3b and Supplementary Fig. 19). These results indicate a positive effect of G196T on $Ca$PETase[WT] was applied similarly to $Ca$PETase[M8].

$Ca$PETase[M9] showed much higher activity at all temperature conditions from 30 to 70 °C, and particularly, showed 41.7-fold higher specific activity at 60 °C than $Ca$PETase[WT] (Supplementary Fig. 20). The result was also reproduced in a scale-up system of 50-mL shaking flasks (Supplementary Fig. 21). These results demonstrated the improved enzyme activity and reinforced thermostability of $Ca$PETase[M9]. The improved thermostability of the variant was further verified through heat inactivation experiments, where $Ca$PETase[M9] maintained its activity even after incubation at 60 °C for 12 h, whereas $Ca$PETase[WT] showed complete loss of activity within an hour (Supplementary Fig. 22).

We then compared the PET hydrolytic activity of $Ca$PETase[M9] with LCC[ICCG] towards PC-PET and Cry-PET at temperatures ranging from 30 to 60 °C. $Ca$PETase[M9] showed significantly higher PET hydrolytic activity compared to LCC[ICCG], at 30 °C and 40 °C (Fig. 3c). At 50 °C and 60 °C, $Ca$PETase[M9] showed quite similar activity compared with LCC[ICCG] (Fig. 3c).

## Structural insights into the enhanced PET degrading capacity of $Ca$PETase[M9]

To provide structural insights into the enhanced PET-degrading capacity of $Ca$PETase[M9], we determined its crystal structure at a resolution of 1.53 Å (Fig. 4 and Supplementary Table 2). The formation of the introduced DS2 and DS4 disulfide bonds was clearly observed in $Ca$PETase[M9], and the S−S interatomic length of both disulfide bonds was within the optimal disulfide bond length range (Fig. 4 and Supplementary Fig. 23). Interestingly, DS4 was located in the vicinity of one of the calcium binding sites of Cut190 and the mutation point of $Is$PETase R280A[42,43], and the formation of DS4 also caused significant changes in the surface electrostatic potential and neighboring region conformation (Fig. 4 and Supplementary Figs. 23 and 24). The side chain of the mutated V129T was flipped to form a hydrogen bond with the adjacent T131 and D132 residues, thereby further stabilizing the β4−α3 connecting loop (Fig. 4 and Supplementary Figs. 23 and 25). With respect to R198K, the mutated lysine residue moved inward to form hydrogen bonds with the main chains of N195 and D222, thereby stabilizing the wobbly tryptophan-containing loop (Fig. 4 and Supplementary Fig. 23). The mutated G196T formed a water-mediated hydrogen bond with the adjacent N195 residue, resulting in further stabilization of the wobbly tryptophan-containing loop (Fig. 4 and Supplementary Fig. 23). The A155R mutation changed the hydrophobic surface to a positive charge, which seems to increase the attachment of the enzyme to the PET surface, as suggested by a previous report (Fig. 4 and Supplementary Fig. 23)[44]. Finally, the N109A mutation appeared to strengthen internal hydrophobic interactions (Fig. 4 and Supplementary Figs. 23 and 26).

## Evaluation of $Ca$PETase[M9] in a pH-stat bioreactor

To evaluate the industrial applicability of $Ca$PETase[M9], we conducted a PET decomposition experiment in a pH-stat bioreactor using PC-PET[Transparent] as a substrate (Supplementary Fig. 27). The bioreactor was operated at 55 °C using 2.70 mg$_{enzyme}$ · g$_{PET}^{-1}$, and the pH was continuously titrated at 8.0 by adding NaOH. The decomposition rate was measured by monitoring released amounts of MHET and TPA. After a short lag phase of an hour, which was required for initial hydrophilization, the PET degradation rate increased exponentially, and 50% of PC-PET[Transparent] was depolymerized within 4 h (Fig. 5a). In the second half of the reaction, the degradation rate slightly decreased because of a decrease in the amount of substrate; however, a final degradation rate of 94.1% was achieved after 12 h (Fig. 5a). This was a significant result in terms of showing that a high depolymerization rate of 90% or more could be achieved even at 55 °C, which is a temperature condition relatively lower than the $T_g$ temperature. This result also indicated that $Ca$PETase[M9] has significant PET hydrolytic activity and thermostability comparable to other benchmark biocatalysts. We also performed decomposition of a post-consumer colored PET powder (PC-PET[Colored]), which is known to be relatively difficult to recycle because of the presence of colors, additives, multilayer structure, labels and other complexities[45]. Interestingly, the depolymerization rate of PC-PET[Colored] was almost identical to that of PC-PET[Transparent], showing 50% depolymerization within 4 h (Fig. 5b); however, the final depolymerization rate of PC-PET[Colored] was 89.2% at 12 h, which was slightly lower than that of PC-PET[Transparent] (Fig. 5b). This is probably due to the impurities present in PC-PET[Colored]. These results demonstrate that unlike other recycling methods, biorecycling of PET plastic can be achieved regardless of the color of PET plastic.

Finally, we determined whether untreated post-consumer PET containers can be depolymerized by $Ca$PETase[M9]. As depolymerization proceeded, the PET film became opaque and thin, and the PET film disappeared completely in 3 days (Fig. 5c). These results suggest that $Ca$PETase[M9] can be utilized for decomposing PET plastics with various physical properties.

## Discussion

As plastic waste is emerging as an environmental issue, research on enzymatic PET depolymerization for plastic waste is expanding as a sustainable alternative for plastic waste treatment and recycling. Therefore, research focused on the discovery of promising PET-degrading enzymes has been actively conducted, and to date, more than 20 PET-degrading enzymes have been biochemically identified and characterized[46,47]. Moreover, studies on developing robust PET-degrading enzymes through various engineering strategies, such as rational designing, directed evolution, and computation-guided approaches, have been conducted. Although reports on the development of efficient PET-degrading enzyme variants are rapidly increasing, the target enzymes for the development of superior variants are focused on a few PET hydrolases, such as $Is$PETase and LCC, which have excellent catalytic and thermostability properties, respectively. In this work, we reported a PET-degrading enzyme, $Ca$PETase, with both mesophilic and thermophilic PET hydrolase properties; It exhibits higher activity than $Is$PETase under the ambient conditions and has a $T_m$ value of 66.8 °C. A positioning of serine for tryptophan wobbling and an extended loop for enzyme-substrate access were proposed as key structural features of $Is$PETase for its high activity at ambient temperature[21,22,48]. Although $Ca$PETase shows high activity at ambient temperature, the enzyme does not share these structural features of mesophilic PET hydrolase (Supplementary Fig. 7). Moreover, its backbone fluctuation profile showed that overall flexibility at the active site is lower than that of $Is$PETase (Fig. 2b), suggesting that it has

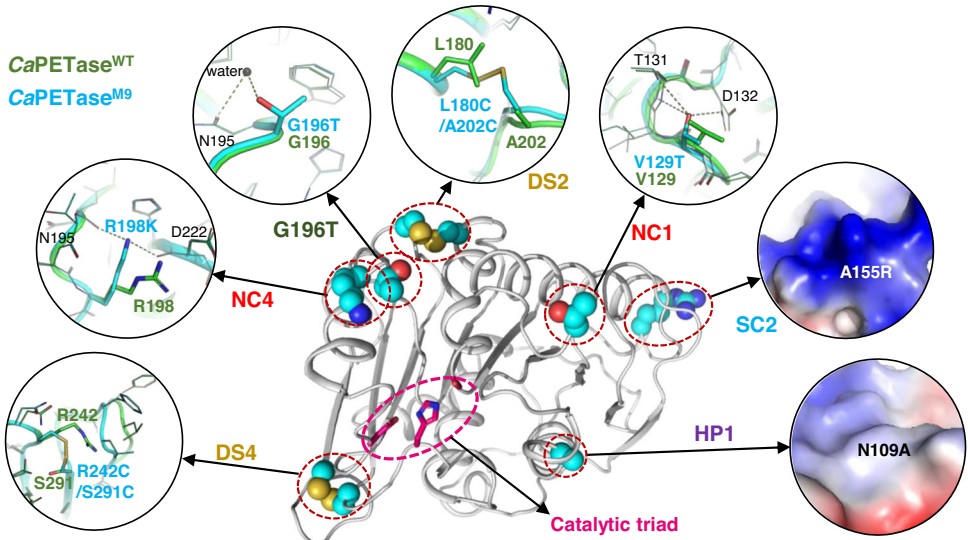

**Fig. 4 | Structural analysis of the mutated residues of _Ca_PETase^M9.** The crystal structure of _Ca_PETase^M9 is shown as a cartoon diagram, and the mutated residues are shown as a stick or a surface electrostatic potential model.

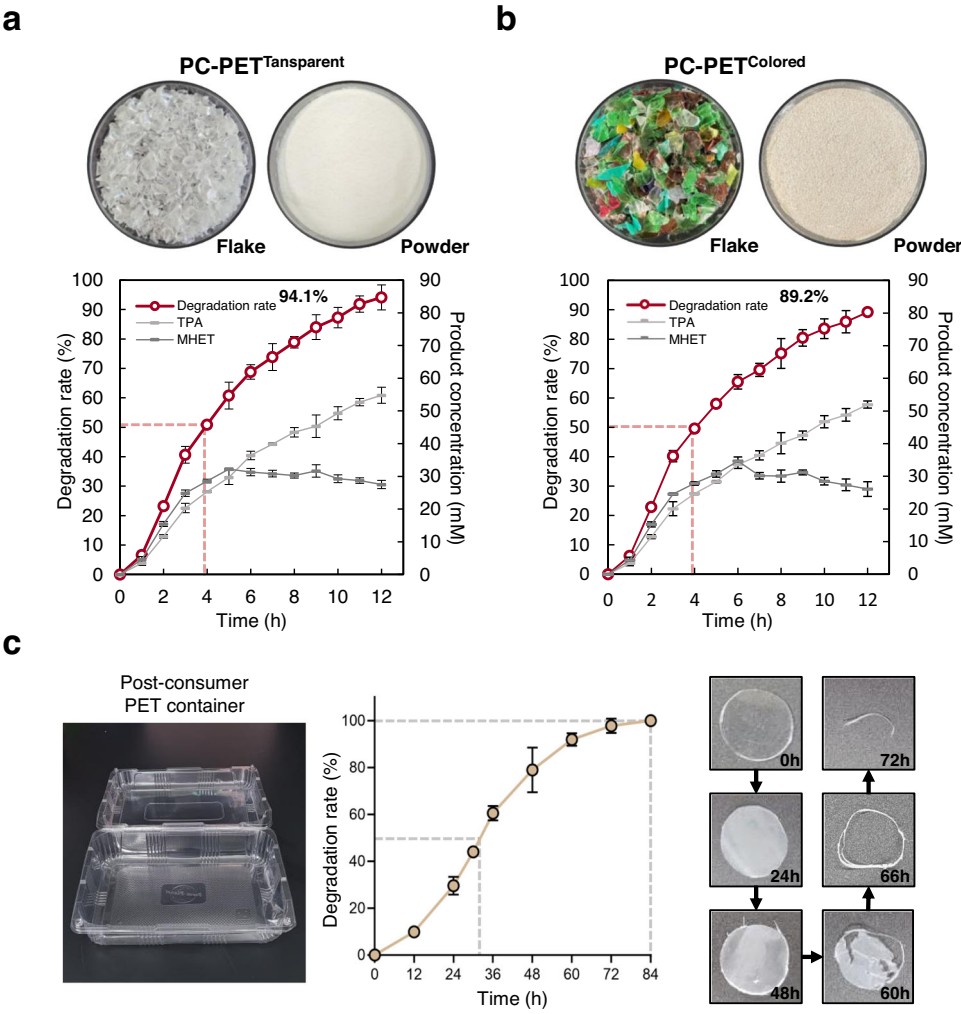

**Fig. 5 | Evaluation of _Ca_PETase^M9 in a pH-stat bioreactor.** Decomposition of post-consumer transparent PET powder (PC-PET^Transparent) (**a**) and post-consumer colored PET powder (PC-PET^Colored) (**b**) in a pH-stat bioreactor using _Ca_PETase^M9. Reactions were performed in triplicate independently; Data are presented as

mean values ± SD. **c** Complete degradation of a post-consumer PET container using _Ca_PETase^M9 at 60 °C. Reactions were performed in triplicate; Data are presented as mean values ± SD.

high activity at ambient conditions through a structural feature different from that of *Is*PETase. Structural analysis of *Ca*PETase with three representative PET-degrading enzymes, such as *Is*PETase, *Tf*Cut2, and LCC, showed that *Ca*PETase has unique backbone torsions of the active site loops caused by three major internal networking differences. One of the distinct internal networks is located under the β8−α5 loop (Supplementary Fig. 13). It is noteworthy to note that the unique disulfide bond of *Is*PETase corresponds to this region and that replacements of the two cysteine residues with alanine severely decrease its BHET and PET hydrolytic activity[22]. Compared with LCC and *Tf*Cut2 having a A-A-T network, *Ca*PETase has a unique G212-F248-A192 network having the bulky hydrophobic side chain 'anchoring' to the core region (Supplementary Fig. 13). Presumably, the anchoring phenylalanine in the network allowed *Ca*PETase to maintain lower perturbation of the catalytic H246 position in the front region of β8−α5 loop, although the enzyme shows a relatively higher backbone fluctuation at the middle region of the β8−α5 loop (Fig. 2b, c). The middle region of the β8−α5 loop in *Ca*PETase contains a unique T250 residue, which interacts with W168 and the β3−α1 loop (Fig. 2b, c). Formation of this unique environment extends to the unique W105-L108-G124 network at the interface between β3−α1 and β4−α2 loops (Supplementary Fig. 12). The R176-W200-F209 network of β6−β7 loop is also noteworthy, as this region has been reported to be important for the activity of mesophilic enzymes[41,42,48]. Taken together, the unique structural features and backbone dynamics profile of *Ca*PETase seem to contribute to its stability and high activity at the same time.

Through the rational design of *Ca*PETase, combinatorial scaffold engineering was performed to increase thermal stability and catalytic efficiency of enzyme, and as a result, considerable improvement was carried out at the $T_m$ value ($\Delta T_m = 16.7\,°C$) and catalytic efficacy (increased by 41.7-folds). As part of further efforts to assess the capacity of *Ca*PETase^M9, comparisons with benchmark enzyme and pH-stat bioreactor operation elucidated that *Ca*PETase^M9 has sufficient PET decomposition performance for industrial applications. Thus, through structure-guided rational protein engineering of *Ca*PETase, we developed *Ca*PETase^M9 with dramatically enhanced PET hydrolytic activity and thermostability and demonstrated that this variant has sufficient PET depolymerization ability for industrial applications. Our engineering strategy could provide insights for the improvement of catalytic activity and thermostability for other enzymes along with the rational design used in this study. The emergence of a *Ca*PETase having both mesophilic and thermophilic PET hydrolase properties will broaden the spectrum of enzyme for enzyme engineering and accelerate the implementation of biological recycling of PET.

## Methods

### Preparation of PET samples

Post-consumer transparent PET powder (PC-PET^Transparent) and post-consumer colored PET powder (PC-PET^Colored) used in this study was produced by grinding PET flakes purchased from a recycling company (Yoochang R&C, Republic of Korea). Prior to milling to produce PET powder, the PET flakes were washed with water and dried at a temperature of 160 °C for 6 h. The extensively dried PET flakes were fed into the twin-screw extruder with set temperatures to 260 °C in the extruder zones, 275 °C in the discharge zones at 275 °C, and 275 °C in the die plate, and an extrusion speed of 50 rpm. The filament was then sequentially cooled into two water baths at temperatures of 75 °C and 60 °C, and excess water was removed using air wipers. Then, a single hole die with a diameter of 3.0 mm was employed for pelletizing, resulting in pellets of approximately 3 mm in length. The obtained PET pellets were micronized using a cryogenic Micro Ball Mill. Particle size distribution analysis of the PET samples was conducted using a HORIBA LA 960S2 particle size analyzer in accordance with ISO 13320. The particle size distributions, crystallinities, and molecular weights

(Mn, Mw, Mz) of PC-PET^Transparent and PC-PET^Colored are presented in Supplementary Figs. 28, 29 and Supplementary Table. 3.

### Selection of PETase candidates and phylogenetic analysis

Sequences were retrieved from the NCBI nonredundant (nr) protein sequence database via PSI-BLAST (parameters: E-value of 1.00E-58, % identity >40%) using *I. sakaiensis* PETase (Accession code: GAP38373.1) to query the model. Partial genes were removed from the retrieved sequences, and then sequences of the PETase candidates were randomly selected from the dataset. Multiple sequence alignment was performed using Clustal omega[49]. Phylogenetic reconstruction was performed via maximum likelihood (ML) using MEGA11 with the Dayhoff w/freq. model[50,51]. The tree with the highest log likelihood (−10401.10) is shown. Initial trees for heuristic search were obtained automatically by applying Neighbor-Join and BioNJ algorithms to a matrix of pairwise distances estimated using the JTT model, and then the topology with a superior log likelihood value was selected. Discrete gamma distribution was used to model evolutionary rate differences among sites (5 categories [+G, parameter = 1.6961]). The rate variation model allowed for some sites to be evolutionarily invariable ([+I], 4.28% sites). This analysis involved 27 amino acid sequences, and there was a total of 444 positions in the final dataset. Bootstrap values were obtained from 1000 replicates[52]. Sequence alignment of enzymes used in phylogenetic tree analysis was depicted as a graphical illustration using ESPript3 (Supplementary Fig. 1)[53]. Pairwise identity and similarity of the proteins were calculated using the Clustal omega method, and a sequence identity matrix was created using excel software.

### Plasmid construction and mutagenesis

All proteins used in this study were codons optimized for *E. coli* expression. The signal peptides sequences were removed, and NdeI and XhoI restriction sites were added to the 5′ and 3′ ends of the target gene sequences, respectively. These DNA sequences were synthesized by Twist Bioscience without signal peptides sequences. The signal peptide cleavage sites for each sequence were predicted using SignalP 3.0 server. Ten PETase candidate genes as well as *Tf*Cut2 and LCC were subcloned into a pET21b expression vector, and *Is*PETase was subcloned into a pET15b expression vector. Site-directed mutagenesis experiments were performed via polymerase chain reaction (PCR) with the oligonucleotides listed in Supplementary Table 4. All variants were verified by sequencing.

### Recombinant protein expression and purification in *E. coli*

The constructed vectors were transformed into *E. coli* strain Rosetta gami-B (DE3), which was grown in lysogeny broth containing 100 mg L⁻¹ ampicillin at 37 °C until the optical density of the culture broth reached a value of 0.6 at 600 nm. After induction by the addition of 0.5 mM isopropyl β-D-1-thiogalactopyranoside, the culture was further incubated at 18 °C for 20 h. Cells were harvested via centrifugation at 4000 × g for 15 min at 20 °C. The cell pellet of the 10 PETase candidates, LCC, and *Tf*Cut2 were resuspended in lysis buffer A (40 mM Tris-HCl, pH 8.0), and the cell pellet of *Is*PETase was resuspended in lysis buffer B (50 mM Na₂HPO₄-HCl, pH 7.0). Cell lysis was performed via ultrasonication, and the cell lysates were centrifuged at 13,500 × g for 30 min. The cell-free crude enzymes were purified using a Ni-NTA agarose column (Qiagen). After washing with buffers A and B containing 30 mM imidazole, the target proteins were eluted using 300 mM imidazole in buffers A and B. All protein purification experiments were performed at 4 °C. The purified proteins were verified via sodium dodecyl sulfate polyacrylamide gel electrophoresis and quantified using a BioTek™ Epoch microplate spectrophotometer and Gen5™ microplate data analysis software. The purified enzymes were concentrated using an Amicon Ultra Centrifugal filter unit (molecular weight cutoff value of 10,000, Millipore, Billerica, MA, USA).

The expression and purification of the variants of CaPETase were performed under the same conditions as the wild-type.

## Melting temperature ($T_m$) measurements

The thermal stability of the enzymes was determined by establishing melting curves with a protein thermal shift dye (Applied Biosystems) using a StepOnePlus Real-time PCR instrument (Thermo Fisher Scientific). The reaction mixture contained 5 μg of proteins, 100 mM $Na_2HPO_4$-HCl (pH 7.0), and 20 μL of 1× protein thermal shift dye. Signal changes representing protein denaturation were monitored by increasing the temperature from 25 °C to 99 °C. Melting temperatures were estimated from the first derivative curve. The first derivative curves of RFU for CaPETase$^{WT}$ and its variants are presented in Supplementary Fig. 30.

## Crystallization and structure determination of CaPETase$^{WT}$ and CaPETase$^{M9}$

Crystallization of CaPETase$^{WT}$ and CaPETase$^{M9}$ was initially performed with commercially available spares-matrix screens Wizard I and II (Rigaku), PEG/Ion, Index (Hampton Research), and Wizard CRYO I and II (Rigaku) using the sitting-drop vapor diffusion method at 20 °C. For each experiment, 1.0 μL of the protein solution (CaPETase$^{WT}$: 18.4 mg mL$^{-1}$ in 40 mM Tris, pH 8.0, and CaPETase$^{M9}$: 22.3 mg mL$^{-1}$ in 40 mM Tris, pH 8.0) was mixed with 1.0 μL of reservoir solution and the drop was equilibrated against 50 μL of reservoir solution. The crystal of CaPETase$^{WT}$ appeared under conditions of 12% (v/v) PEG 3350 and 0.1 M sodium malonate (pH 4.0). The crystal of CaPETase$^{M9}$ was formed under conditions of 10% (v/v) polyethylene glycol monomethyl ether 5,000, 0.1 M HEPES, pH 7.0, and 5% tacsimate, pH 7.0. The highest quality crystals of CaPETase$^{WT}$ and CaPETase$^{M9}$ were transferred into a cryoprotectant solution composed of the corresponding conditions described above with 25% (v/v) glycerol. The crystals were fished out with a loop and flash-frozen with a spout of liquid nitrogen. Data were collected using a 7 A beamline of the Pohang Accelerator Laboratory (Republic of Korea) under 100 K[54]. All diffraction data were indexed, integrated, and scaled using HKL2000 software[55]. The CaPETase$^{WT}$ crystals belonged to a space group $P2_12_12$ with a unit cell parameter: $a = 82.31$ Å, $b = 82.45$ Å, $c = 87.39$ Å, and $\alpha = \beta = \gamma = 90°$. The two CaPETase$^{WT}$ molecules were in an asymmetric unit, and the crystal volume per unit of protein was 2.68 Å$^3$ Da$^{-1}$, corresponding to a solvent content of 52.25%[56]. The structure of CaPETase$^{WT}$ was determined via molecular replacement with the CCP4 version of MOLREP using the structure of cutinase 1 from Thermobifida cellulosilytica (PDB code: 5LUI) as a search model[57,58]. The structure model was built using the WinCoot program, and structure refinement was performed using the CCP4 REFMAC5 program[59,60]. The crystal of CaPETase$^{M9}$ belonged to the space group $P2_12_12$ with a unit cell parameter: $a = 41.02$ Å, $b = 112.31$ Å, $c = 112.54$ Å, and $\alpha = \beta = \gamma = 90°$. The asymmetric unit contained two molecules of CaPETase$^{M9}$, and the crystal volume per unit of protein was 2.3468 Å$^3$ Da$^{-1}$, corresponding to a solvent content of 45.37%[56]. The structure of CaPETase$^{M9}$ was determined via molecular replacement with the CCP4 version of MOLREP using the structure of CaPETase$^{WT}$ as a search model[57]. Model building and refinement were performed as described above. The statistics are summarized in Supplementary Table 2. The structures of CaPETase$^{WT}$ and CaPETase$^{M9}$ were deposited in the Protein Data Bank, with PDB codes 7YM9 and 7YME, respectively.

## MD simulations

The deposited structures of 5XJH (G30-S290), 4EB0 (S36-Q293), 4CG1 (A1-F261), and 7YM9 (D43-F299) were used for IsPETase, LCC, TfCut2, and CaPETase residues and inner cavity water coordinates. All protein models were prepared with protonation states at pH 9, as calculated by propka3[61]. The systems were prepared with ff19SB and OPC parameters for protein and explicit water and solvated in a cubic box of

70 Å length with 0.1 M Na$^+$ and Cl$^-$ ions. We minimized the systems using gromacs with conjugate gradient algorithm, and NVT-equilibrated for 1 ns at 323.15 K, and then NPT-equilibrated for 1 ns with a force constraint of 1000 kJ·mol$^{-1}$·nm$^{-2}$. Production simulations were performed for 200 ns at 323.15 K, and rms fluctuations were calculated with 10 ps interval trajectories. All simulations were performed using Gromacs 2021.3 version package[62].

## PET degradation assays

In the preliminary PET degradation assay of eight PETase candidates, PC-PET$^{Transparent}$ (15 mg mL$^{-1}$) was soaked in 50 mM Glycine-NaOH pH 9.0 buffer with 500 nM enzyme. Cry-PET (15 mg mL$^{-1}$) and hole-punched amorphous PET film (⌀6 mm, roughly 10 mg) were soaked in 50 mM Glycine-NaOH pH 9.0 buffer with 2 μM enzyme. The reaction mixtures (1 mL) were then incubated at temperatures of 30 °C, 40 °C, 50 °C, and 60 °C for 3 days. The reaction was terminated by heating the mixture at 95 °C for 10 min. The reaction mixture with PET powder was filtered using a PVDF syringe filter (0.22 μm) to eliminate the residual substrate. The samples were then analyzed using HPLC. All experiments were performed in triplicate. The pH profiles on eight PETase candidates were performed using PC-PET$^{Transparent}$ (15 mg mL$^{-1}$) in 50 mM sodium phosphate buffer (pH 6-7), 50 mM Tris-HCl buffer (pH 7-9), and 50 mM Glycine-NaOH buffer (pH 9-10) at 30 °C for 3 days. All experiments were performed in triplicate. To compare the PET hydrolytic activity of CaPETase with LCC, IsPETase, and TfCut2, 15 mg of Cry-PET was prepared and submerged in 1 mL of 50 mM glycine-NaOH pH 9.0 buffer with 2 μM enzyme. The reaction mixture was incubated at 30 °C, 40 °C, 50 °C, and 60 °C for 12 h. Then, the reaction mixture was treated as described above and analyzed using HPLC. All experiments were performed in triplicate.

The PET hydrolytic activity of CaPETase$^{WT}$ and its variants was determined using PC-PET$^{Transparent}$. Enzyme reactions were performed with 500 nM enzymes in 1 mL of 50 mM glycine-NaOH (pH 9.0) buffer at 40 °C for 24 h. The reaction mixture was treated as described above and analyzed using HPLC. All experiments were performed in triplicate. To evaluate the PET-hydrolyzing activity of the combinatorial variants of CaPETase, 500 nM enzyme was incubated with 100 mM glycine-NaOH (pH 9.0) and 15 mg of PC-PET$^{Transparent}$ at 40 °C for 24 h and 60 °C for 6 h, respectively. The reaction mixture was treated as described above and twofold diluted samples were analyzed using HPLC. All experiments were performed in triplicate.

The comparison of PET hydrolytic activities of CaPETase$^{WT}$ and CaPETase$^{M9}$ were performed using PC-PET$^{Transparent}$ (15 mg mL$^{-1}$) at 30, 40, 50, 60, 70 °C with 1 μM enzymes in 1 mL of 50 mM glycine-NaOH (pH 9.0) buffer. To further evaluate CaPETase$^{M9}$ for PET depolymerization, the enzyme (1 μM) was added to 50 mL of 200 mM Glycine-NaOH pH 9.0, containing 100 mg of PC-PET$^{Transparent}$ in a 250-mL flask. The flask was incubated in a shaking incubator at 40 °C, 50 °C, 60 °C and 180 rpm (40 °C for 72 h; 50 °C for 48 h; and 60 °C for 12 h). The reaction mixtures were sampled at different time points to quantify the total amount of released PET monomers. The samples were filtered as described above and 4-fold diluted samples were analyzed using HPLC. All experiments were performed in triplicate.

To compare the PET hydrolytic activity of CaPETase$^{M9}$ with LCC$^{ICCG}$, a membrane enclosed enzymatic catalysis (MEEC) system were used. In this system, the PET degradation reaction was induced on a cellulose dialysis membrane (Spectra/Por 2 Trial Kit, 12–14 kD, flat width 25 mm; Repligen, USA), and PC-PET$^{Transparent}$ and Cry-PET were used as substrate. 50 (± 2) mg of PC-PET$^{Transparent}$ and Cry-PET substrate were immersed in a 4 mL of 200 mM Glycine-NaOH pH 9.0 with 1 μM and 4 μM enzyme, respectively. Each reaction was packaged in a 4-mL dialysis bag was immersed in 200 mL of 200 mM Glycine-NaOH pH 9.0. The reaction was performed at 30 °C, 40 °C, 50 °C, and 60 °C with shaking at 80 rpm in a small shaking incubator (JEIO TECH, Republic of Korea). Sample was taken from the external equilibrium solution,

diluted 5-fold, and then analyzed by HPLC. All experiments were performed in triplicate.

### Enzyme heat-inactivation

Protein heat inactivation experiments were performed using CaPETase[WT] and CaPETase[M9]. Enzyme mixtures (500 nM enzyme in 1 mL of 50 mM glycine-NaOH, pH 9.0) were incubated at 60 °C and 70 °C for 10 min to 12 h and 5 min to 12 h, respectively. Then, 15 mg of PC-PET[Transparent] was added to the heat-inactivated mixture, and the reaction mixture was incubated at 40 °C for 1 day. The mixture was filtered using a PVDF syringe filter (0.22 μm) and the samples were analyzed using HPLC.

### PET depolymerization in a pH-stat bioreactor

A customized stirred bioreactor (MARADO-PDA, BIOCNS Co. Ltd, Korea) equipped with two standard Rushton impellers was designed for PET depolymerization. It was interfaced with a personal computer for data recording. The temperature was regulated using a refrigeration and heating bath circulator (RW3-1025, JEIO TECH, Daejeon, Republic of Korea). CaPETase[M9] (10.2 mg) was added to a 150-mL solution containing 10 mM $Na_2HPO_4$ (pH 8.0) and 3.75 g PC-PET and incubated at 55 °C and 200 rpm. The pH was regulated at 8.0 by the adding 0.3575 M NaOH (40%; extra pure grade, Duksan, Republic of Korea) solution using a PID controller. The reactor solution was sampled during the reaction and filtered through a PVDF syringe filter (0.22 μm) to remove residual PET powder. The resulting samples were diluted with 50 mM glycine-NaOH (pH 9.0) and analyzed using HPLC to quantify the amount of released PET monomer. The PET depolymerization rate was calculated by considering the sum of the hydrolysis products (TPA and MHET). All experiments were performed in triplicate.

### Depolymerization of post-consumer PET container

Fragments of a post-consumer PET container (salad container) were prepared in a circular form (⌀14 mm, 50 mg). The reaction was performed using a membrane-enclosed enzymatic catalysis method. The dialysis membrane bag (Carl Roth, Product number 0653.1, MWCO 14000) was loaded with 1 μM of CaPETase[M9] dissolved in 4 mL of 50 mM glycine-NaOH (pH 9.0) buffer and post-consumer PET fragment. The filled membrane bag was immersed into a container containing 196 mL of 50 mM glycine-NaOH (pH 9.0) buffer and incubated on a shaker for 84 h at 60 °C and 80 rpm. Samples were removed from the solution on the outside at different time points (0, 12, 24, 30, 36, 48, 60, 72, and 84 h) to quantify the total amount of released PET monomers. The samples were analyzed using HPLC. All experiments were performed in triplicate.

### HPLC analysis

Samples were analyzed using a CBM-20A (Shimadzu, Kyoto, Japan) connected to a UV/Vis detector (SPD-20A) and a C18 column (Shimadzu Shim-pack GIS ODS-I C18, 5 μm, 4.6 × 150 mm). Mobile phases A (0.1% formic acid) and B (20% acetonitrile) were used at a flow rate of 1 mL min$^{-1}$. The elution program was as follows: 0–2 min, 60%–80% buffer B, linear gradient; 2–8 min, 80%–100% buffer B, linear gradient; 8–16 min, 100% buffer B. The hydrolysis products (TPA, MHET, and BHET) were separated at 40 °C and detected at 260 nm. Amounts of TPA, MHET, and BHET were measured using standard curves prepared from commercial TPA (Sigma-Aldrich, MO, USA, Cat. No.: 185361), BHET (Sigma-Aldrich, MO, USA, Cat. No.: 465151), and MHET (Ambeed, IL, USA, Cat. No.: A875019).

### Differential scanning calorimetry for crystallinity measurement of PET samples

The thermal crystallinity of PET samples was estimated using a differential scanning calorimetry (DSC) instrument (Q2000, TA instrument). For the DSC thermographs, 5 mg of samples were equilibrated at 30 °C and heated to 300 °C at a heating rate of 10 °C min$^{-1}$. The sample was quenched to 30 °C at 10 °C min$^{-1}$. Heat of fusion (ΔHm) and cold crystallization (ΔHc) were determined by integrating the areas (J g-1) under the peaks. The enthalpy of melting for a fully crystalline PET (ΔHf) was 140.1 J/g. The percent of crystallinity was calculated based on the following equation:

$$crystallinity(\%) = \frac{(\triangle Hm - \triangle Hc)}{\triangle Hf} \times 100\% \qquad (1)$$

DSC curves for the crystallinity of the PET samples are presented in Supplementary Fig. 29.

### Gel permeation chromatography (GPC)

The molecular weight of the PET samples was measured using gel-permeation chromatography (GPC) using an EcoSEC HLC-8320 GPC instrument (Tosoh Bioscience, Tokyo, Japan). The TSKgel SuperAWM-H column was used with a mobile phase of 0.01 N sodium trifluoroacetate (NaTFA) in hexafluoroisopropanol (HFIP) at a flow rate 0.3 mL min$^{-1}$ at 40 °C, and 30 min run time per sample. The injected sample volume was 20 μL. The molecular weights (Mn, Mw, Mp) were determined relative to polymethylmethacrylate (PMMA) standards. The PET samples and PMMA were dissolved in 6% v/v hexafluoroisopropanol (HFIP) in chloroform to a concentration of 3 mg mL$^{-1}$, and the solution was filtered through a polytetrafluoroethylene (PTFE) membrane with a pore size of 0.45 μm.

### Reporting summary

Further information on research design is available in the Nature Portfolio Reporting Summary linked to this article.

## Data availability

The refined models of CaPETase[WT], CaPETase[M9] generated in this study have been deposited in Protein Data Bank with PDB codes 7YM9 and 7YME. We have used the following published structure, PDB code 5LUI, as a search model for the initial molecular replacement. The raw data for figures generated in this study are provided in the Source Data file. The percent identity values for sequences used in phylogenetic analysis were provided in the Supplementary Fig. 31. The data that support the findings of this study are available from the corresponding author upon request. Source data are provided with this paper.

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

## Acknowledgements

This research was supported by the Bio & Medical Technology Development Program of the National Research Foundation (NRF) funded by the Ministry of Science & ICT (NRF-2020M3A9I5037635) and the Cooperative Research Program for Agricultural Science & Technology Development (Project No. PJ01492602), Rural Development Administration, Republic of Korea. This research was also supported by the Technology Innovation Program (20018132, Development of the biodegradable polybutylene plastics, PBAT and PBS, from biomass) funded By the Ministry of Trade, Industry & Energy (MOTIE, Korea).

## Author contributions

K.-J.K. conceived the project and supervised the research. H.H. and D.K. performed the crystallographic experiment and S.H. contributed to the computational analysis. H.H. and D.K. designed the enzyme variants and performed the PET degradation experiments. H.H., D.K., and J.P. analyzed the data. H.H. and K.-J.K. participated in the manuscript writing. All author discussed results and S.H. and J.J. provided guidance of this manuscript.

## Competing interests

The authors declare no competing interest.
