## [Peer Review File · Nature Communications]

REVIEWER COMMENTS

Reviewer #1 (Remarks to the Author):

From a modest collection of PETase candidates, Hong et al. find PET hydrolase from *Cryptosporangium aurantiacum* (CaPETase) has the highest PET degradation activity across the selected enzymes. They solve the structure of the CaPETase and show how it biochemically react with PET substrate. Based on this, They further constructed a CaPETaseM9 variant that exhibits robust thermostability and 41-fold enhanced PET hydrolytic activity compared with CaPETaseWT. They show that CaPETaseM9 is a potent bioreactor for PET recycling. The authors claim that CaPETase is an outstanding template PETase for further engineering and repeatedly emphasized that CaPETase has unique structural features which are distinct from other known PETases. However, a thorough characterization of how CaPETase and CaPETaseM9 interact with substrate (via complex structure solving and molecular dynamic simulation), and its possible substrate recognition specificity would be necessary to show how significant its discovery might be as a tool for plastic depolymerization and industrial applications. The claim that CaPETase is a "novel" PET hydrolase is overstated, especially given the well-aligned overall structure superpositions with very small RMSD values and highly conserved enzymatic mechanism with other published PETase such as LCC, IsPETase, TfCut....I believe the real novelty of CaPETase comes from the identification of its better performance at ambient temperature, however, it should be noted that the activity of this enzyme, including its mutants, is still lower than that of LCC and some engineered LCC and IsPETase at extremely high temperatures. Given this reasoning, I don't think this paper has the novelty and impact I expect. Nevertheless, I still hope that the author will enhance the appeal of this paper by strengthening the description and discussion of the protein engineering section. Furthermore, I have several concerns which are detailed below.

1. What rationale did the author introduce the additional disulfide bonds? With reference to the successful engineering works on existing PETases such as LCC and IsPETase? Or based on different strategies? Since all of the following combinatorial mutations are based on DS2-DS4, have the authors summarized the different effects of disulfide bonds at different positions on activity and thermal stability?
2. Based on existing literature, different PET hydrolase may have different optimal reaction temperature, and some enzyme activities even show responses to different metal ions. Therefore, it is necessary to test CaPETase activity at different temperatures and with different ions.
3. By analyzing the structure, the author found that "CaPETase also showed significant differences in the backbone torsion angles at the five connecting loops ($\beta 3-\alpha 1$, $\beta 4-\alpha 2$, $\beta 6-\beta 7$, $\beta 7-\alpha 4$, and $\beta 8-\alpha 5$) that form an active site, whereas comparisons of the corresponding loops between the other three PET hydrolases exhibited less differences", Indeed, the CaPETase structure compared with other three PET hydrolases exhibited very limited differences, most of the point mutations designed by the author are around active site via sequence alignment, rather than the so-called unique structural characteristics.
4. In line 191-195, "CaPETase also contains a unique I102 residue in the $\beta 3-\alpha 1$ loop showing the largest torsion differences, whereas other PET hydrolases contain a highly conserved threonine residue at the corresponding position, which probably enables CaPETase to form a relatively wider substrate binding cleft (Fig. 3d, e and Supplementary Fig. 11)." The author could try several more substrates to verify whether the enzyme has advantages in substrate selectivity, or discuss it in detail in the discussion.
5. Compared with CaPETaseWT, the CaPETaseM9 does show great advantages in hydrolytic activity and thermostability. However, the author should highlight the advantages of this rational engineering mutant in the discussion and compare it with other enzyme mutants.
6. The discussion is too brief, and the advantages of CaPETase over IsPETase, Tfcutinase, and LCC are not discussed in depth, nor are the reasons for this advantage analyzed in terms of which structural differences give rise to this advantage. What are the implications for the screening and modification of other related enzymes?

Minor concerns

1. In this manuscript, there are some many formatting errors. For example, "4000g" to "4000 \times g" (line 437), "13500g" to "13500 \times g" (line 441), "mg/ml" to "mg/mL" (line 465), "ml" to "mL" (line 513 and line 514).
2. Please carefully check the references. Such as "line 590", "line 620", and "line 663".
3. In supplementary Fig. 2c, SHM40309.1 had an extremely high PET hydrolytic activity, the label

1193.12 coincides with the vertical axis

4. In supplementary Fig. 4, Please carefully check the listed residues, some of them don't seem to match the sequence.

5. In supplementary Table. 2, please provide CC1/2 values, outer shell values and clash scores in "Refinement". As well, footnotes are also required for parameters such as R/Rfree values etc.

6. In line 214, change "PyMol" to "PyMOL".

7. In line 194, change "Fig. 3d, e" to "Fig. 2d, e".

Reviewer #2 (Remarks to the Author):

The work of Ho et al. provides new results in the field of enzymatic degradation of PET. In this work, a new enzyme from the bacterium *Cryptosporangium aurantiacum* (CaPETase) capable of degrading PET in a wide range of temperatures is characterized, being more active at moderate temperatures than PETase from *Ideonella sakaiensis*, the most active mesophilic enzyme so far. In addition, crystallographic structures of the enzyme are provided showing that CaPETase shows structural differences with other PETases. Furthermore, by combination of point mutations the activity of CaPETase is improved by more than 31-fold at 60°C and the applicability of this new mutant enzyme, CaPETaseM9, in a bioreactor with post-consumer PET is demonstrated. The experimental data is convincing and manuscript is well written and prepared. The data presented in this comprehensive work enriches the scientific field.

However, to improve the quality of an article, I suggest considering the following:

- In the introduction several PET hydrolyzing enzymes are listed, however there is no mention of the enzyme PHL7, which is the most active thermophilic enzyme so far.
- The rationale behind the authors' choice of those ten sequences as PETase candidates is unclear.
- The initial screening of the 10 chosen enzymes solely tests PET hydrolytic activity at 30 °C. If there are any thermophilic enzymes, as in the case of WP 068752972.1, it is not surprising that their activity is very low at 30 °C. It would be interesting to test this enzyme at higher temperatures, closer to its T_m.
- It should be noted that the enzymatic activity comparison between CaPETase, LCC, TfCut2 and IsPETase was conducted at a pH of 9.0 with glycine-NaOH buffer, which is not the optimum pH and buffer system for LCC or TfCut2.
- There are no such figures referred to as "Fig. 3d" "Fig.3e" and "Fig.3f".
- In line 254, the authors said they introduced 6 point-mutations, i.e. DS2, DS4, NC1, NC4, HP1 and SC2. However, as DS2 and DS4 are double mutants, it would be more accurate to say that eight point-mutations were introduced or 6 mutants were created.
- Initial hydrolysis rate values are not provided, but enzyme activity is calculated as the number of products released over a specified period of time. It would be more suitable to provide the values of the initial hydrolysis rates for both the wild-type enzyme and the M9 mutant in order to compare these results with those of other similar enzymes.

Reviewer #3 (Remarks to the Author):

This manuscript describes a novel PET hydrolase and its engineered mutants. The authors determined the crystal structures and analyzed the PET hydrolytic activities in comparison with those of related enzymes. I have some concerns about this manuscript as described below.

1, The results of enzyme activity were shown in product amount, the PET monomer concentration. The product concentration would largely depend on the amount applied to HPLC. The yield estimated from

the substrate, PET, should be described.

2, The PET degradation would largely depend on the glass transition of PET, as reported previously (Nature 580, 216-219, 2020). The authors should describe the temperature dependent activity in correlation with the glass transition temperature.

3, The section "Discussion" seems to be poor. The authors should revise the section, partly rearranged from the section "Results".

Reviewer #4 (Remarks to the Author):

The paper from Kim and colleagues describes discovery and engineering of a new PET hydrolase. The protein engineering here is good and certainly merits publication in a good journal, but there are many problems with the experimental setup, such that the authors will need to do a major overhaul of this work for this study to be publishable in Nature Communications, mostly related to the inability to replicate the results obtained here and the near-complete lack of substrate characterization. There is little-to-no benchmarking here either to existing enzymes (LCC-ICCG would be sufficient in my opinion). The addition of substrate docking and MD simulations would add a lot of value as well in terms of the (rather over-definitive) interpretation of the differences of CaPETase to existing enzymes.

The authors should remove the word "novel" from the title and throughout. By definition, original research should be new and novel. See the editorial from S. Scott and C. Jones in ACS Catalysis for a take on this.

Line 23-24 – the authors seem to suggest that engineering has only been done with LCC and IsPETase. I would remove this or substantially change this. Other enzymes have been and are being engineered. These are just the two most prominent to date.

Line 27-29 – I think that this is a very "hand-wavy" sentence that does not really convey much useful information.

Line 49-52 – the authors should either tone down or remove the qualitative and unfounded statements of "sustainability and ecofriendliness". I would encourage the authors to see Singh et al. Joule 2021 and Uekert et al. Green Chem 2022 on these topics.

Line 62-66 – the authors may want to also see Erikson et al. Nature Comm 2022 and compare their enzyme to the new ones reported therein.

Line 76-77 – see comment above.

Line 100-102 – why not use a standard and commercially available PET substrate for replication of your results? See ref. 8 for a discussion as to why this is essential. Were these experiments benchmarked to any known PET hydrolases on these inaccessible substrates? I would strongly suggest the authors to use commercial substrates that anyone in the world can purchase and that are well characterized and benchmark to known enzymes on those substrates.

Importantly, there is very little mention of substrate characterization here.

Did the authors do this initial screening of 9 new PETases over T and pH? That should be done. See Erickson et al. 2022. This would add a lot of value beyond just CaPETase.

For the sequence homology searches, there are published tools to predict the optimal temperature of an enzyme – did the authors look into this at all?

Line 120-121 – it is not surprising at all that TfCut2 and LCC have low activity at 30°C. This is already known.

Figure 1 – please put the reaction conditions (enzyme mass loading per substrate mass loading, solids

loading, buffer, substrate characteristics, etc.) in the figure caption.

Line 156-157 – since the authors make such a big deal of this difference, why put the images that show this in the SI?

The structural description for the backbone dihedral angles presupposes that the apo-structure solved here is indicative of the PET binding conformation. Were any MD simulations conducted to see the flexibility of the places in the structure where these differences are noted? Did the authors do any intrinsic flexible docking simulations of a PET oligomer?

The structural paragraph repeatedly refers to Figure 3, but I think the authors mean Figure 2.

Figure 2b does not convey much information that is useful.

Figure 2 and the surrounding structural analysis would be greatly strengthened by the addition of intrinsic flexible docking of a PET oligomer into the active site and comparison of that with previously published docking studies.

Reaction conditions should be listed in the Figure 2 caption.

Line 232 – same comment as above – I VERY strongly urge the authors to reconsider their substrate choice and to redo these experiments with substrates that are commercially available and well characterized as a benchmark for these types of experiments.

Line 295-296 – same comments as before – this is nice protein engineering work, but it is not conducted on a substrate that is available for the community, and thus, this work is not reproducible.

Figure 3 – same comments as above; please include the reaction conditions in the caption.

Overall, it would be much better if the authors would report % conversion instead of product amount in all figures. Are the authors only including TPA in the product release? Does this include MHET and BHET? Are those being measured and if so, what is the selectivity to TPA versus BHET and MHET?

How did the authors choose the times they are running the reactions as well? Does the product release plateau or cease at the chosen reaction time(s)? How does the pH change in these screening reactions not done in a pH-controlled system?

Line 341-343 – same comments as above – this work is not reproducible. Please repeat this work with a substrate from e.g. Goodfellow or another commercial vendor and thoroughly characterize the substrate.

Did the authors do a thorough pH/buffer/T screen of the best enzyme and the wild-type enzyme to find the optimal conditions for each? This is not clear to me.

Line 350-352 – I do not know what the authors are trying to convey here. Also, comparing their results to those from Tournier et al. are challenging vis-a-vis substrate inconsistencies.

Line 355 – why is this known? Citation?

Line 359 – are you sure that the differences (which are slight) are because of the impurities? Did you control for particle size, PET MW, crystallinity, etc.? Those factors would have to be exactly controlled for to make this claim.

Line 362 – again, what substrate is this? Is it available for anyone? Is it well characterized?

Line 375-376 – please remove claims of eco friendly and energy-efficient – see analysis studies cited above. This process actually has a long way to go yet to truly be “energy-efficient” and “ecofriendly”.

Methods – Preparation of PET samples – this is not sufficient at all. What is the crystallinity? Mw and

Mn? Particle size distribution? See ref. 8. The authors really ought to use Goodfellow or other chemical vendor-supplied PET. Calling their substrate B-PET is a misnomer. It originated from bottles, yes, but after cryo milling, it will not be bottle-grade PET anymore.

[Authors' point-by-point responses]

Reviewer #1 (Remarks to the Author):

From a modest collection of PETase candidates, Hong et al. find PET hydrolase from *Cryptosporangium aurantiacum* (CaPETase) has the highest PET degradation activity across the selected enzymes. They solve the structure of the CaPETase and show how it biochemically react with PET substrate. Based on this, they further constructed a CaPETaseM9 variant that exhibits robust thermostability and 41-fold enhanced PET hydrolytic activity compared with CaPETaseWT. They show that CaPETaseM9 is a potent bioreactor for PET recycling. The authors claim that CaPETase is an outstanding template PETase for further engineering and repeatedly emphasized that CaPETase has unique structural features which are distinct from other known PETases. However, a thorough characterization of how CaPETase and CaPETaseM9 interact with substrate (via complex structure solving and molecular dynamic simulation), and its possible substrate recognition specificity would be necessary to show how significant it's discovery might be as a tool for plastic depolymerization and industrial applications. The claim that CaPETase is a "novel" PET hydrolase is overstated, especially given the well-aligned overall structure superpositions with very small RMSD values and highly conserved enzymatic mechanism with other published PETase such as LCC, IsPETase, TfCut....I believe the real novelty of CaPETase comes from the identification of its better performance at ambient temperature, however, it should be noted that the activity of this enzyme, including its mutants, is still lower than that of LCC and some engineered LCC and IsPETase at extremely high temperatures. Given this reasoning, I don't think this paper has the novelty and impact I expect. Nevertheless, I still hope that the author will enhance the appeal of this paper by strengthening the description and discussion of the protein engineering section. Furthermore, I have several concerns which are detailed below.

We thank the reviewer for very useful comments.

I understand the reviewer's perspective that the use of the term "novel" may have been overstated and that it should be avoided in the original research paper. So, we removed the word "novel" in this revised manuscript. And I agree that if the mutated our enzyme shows superiority over the improved enzymes reported so far, it would have a much greater impact.

However, what we consider to be important is that the newly discovered CaPETase has intrinsic activity as its starting point that is superior to that of the known enzymes at the ambient temperature. Considering the industrial applications potential of this enzyme and the performance of the enzymes reported so far, the discovery of a novel enzyme with superior activity compared to previously reported enzymes is meaningful in itself. In addition, we performed torsion angle analysis and MD simulation using the CaPETase structure. Through this structural analysis, we aimed to uncover the differences and unique structural properties of CaPETase and gain insights into its characteristics.

As this is still in the early stages of discovery, we believe there is still much room for improvement, just like IsPETase and LCC, which have been developed extensively over the years. Nonetheless, by demonstrating that the engineered enzyme (CaPETase^{M9}) can function sufficiently in a reactor system, we have shown our industrial capabilities, even at this stage. Achieving this level of performance is also valuable, given the performance of newly reported enzymes and improved enzymes that have been considered thus far. We have considered the important comments and suggestions made by the reviewer and have revised the paper accordingly, which has further improved the quality of our work. We are deeply grateful for these valuable comments.

1-1. What rationale did the author introduce the additional disulfide bonds? With reference to the successful engineering works on existing PETases such as LCC and IsPETase? Or based on different

strategies? Since all of the following combinatorial mutations are based on DS2-DS4, have the authors summarized the different effects of disulfide bonds at different positions on activity and thermal stability? [Response] Based on the structure of CaPETase, we have performed a rational design of disulfide bond formation by considering whether the geometric parameters for disulfide bond formation were met and whether the distance between the two residues was appropriate. To prevent reduction of enzyme activity, residues around the catalytic site were excluded. Based on these our analysis, we have manually selected the potential candidates of residue pairs (DS1, DS2, and DS3) as engineering candidates. In case of DS4 (R242C/S291C), we have referred to the LCC ICCG variants described by Tournier et al.. The impact of introducing DS2 and DS4 based on the wild-type on activity and thermal stability was experimentally verified. The effect of disulfide bond formation for each variant was confirmed by conducting activity experiments under conditions that were not greatly affected by temperature and measuring T_m values. Therefore, additional combinatorial engineering was performed based on DS2-DS4, which combines these two points. The effects of DS-2 and DS-4 on each factor (activity, thermal stability) can be seen in Fig. 3a, and an explanation of this is provided in the main text.

1-2. Based on existing literature, different PET hydrolase may have different optimal reaction temperature, and some enzyme activities even show responses to different metal ions. Therefore, it is necessary to test CaPETase activity at different temperatures and with different ions.

[Response]

In case of different temperature, we already tested the activity at different temperature, and the results are described in Figure 1 c, d. It is very important to check the effect of the metal ions on CaPETase. To test the effect of metal ions on CaPETase, we measured the effect on T_m values by addition of three metal ions (Ca²⁺, Mg²⁺, and Mn²⁺), and we could not find any changes on the T_m values. These results indicate that these metal ions do not affect on CaPETase structure and further suggest that the enzyme does not require metal ions. We also measured the activity of CaPETase by addition of different concentrations of Ca²⁺, and the activity was dramatically decreased even by addition of 1 mM Ca²⁺. The results indicate that CaPETase is not a metal ion-dependent enzyme.

The experimental results were shown at the Supplementary Fig. 4, and we also described the fact in the manuscript (line # 126).

1-3. By analyzing the structure, the author found that “CaPETase also showed significant differences in the backbone torsion angles at the five connecting loops ($\beta_3-\alpha_1$, $\beta_4-\alpha_2$, $\beta_6-\beta_7$, $\beta_7-\alpha_4$, and $\beta_8-\alpha_5$) that form an active site, whereas comparisons of the corresponding loops between the other three PET hydrolases exhibited less differences”, Indeed, the CaPETase structure compared with other three PET hydrolases exhibited very limited differences, most of the point mutations designed by the author are around active site via sequence alignment, rather than the so-called unique structural characteristics.

[Response]

This is an important comment. First, we do not think that the difference in backbone torsion angles, which we describe as a unique structural characteristic, is a part that can be verified simply by experiments for any mutations or other experimental validations. The specific regions with differing torsion angles are influenced by the surrounding regions. The torsion angles of a backbone are intrinsic values of protein determined by the entire amino acid sequence and its folding. The backbone torsion angle-based structure alignment is known to have advantages over general 3D structure comparison methods for proteins by comparing backbone topology [*Genomics Inform.* 2011;9:74–78].

In order to further identify the unique structural features of CaPETase, we performed MD simulations of the enzyme and compared with other PET hydrolases. Interestingly, we could observe some differences on local structures, and these differences were highly correlated with the difference in torsion angles. These results further confirm the structural uniqueness of CaPETase.

The results of MD simulations are shown in Figure 2b and described in the manuscript (Line # 196).

Through site-directed mutagenesis experiments, we further investigated the effect of substituting residues in the intrinsic substrate-binding site of CaPETase. This revealed how changes in the substrate-binding cleft could affect enzyme activity and thermal stability. These findings suggest that each enzyme has an optimal substrate-binding site.

1-4. In line 191-195, “CaPETase also contains a unique I102 residue in the β 3- α 1 loop showing the largest torsion differences, whereas other PET hydrolases contain a highly conserved threonine residue at the corresponding position, which probably enables CaPETase to form a relatively wider substrate binding cleft (Fig. 3d, e and Supplementary Fig. 11).” The author could try several more substrates to verify whether the enzyme has advantages in substrate selectivity or discuss it in detail in the discussion. [Response]

Since PET substrates are polymers, it is not easy to accurately validate the impact of the extended substrate binding cleft on PET substrate “selectivity”. To further investigate the enzyme's activity on other substrates, we measured the activity using a monomer substrate, BHET, instead of PET substrate. Our results showed that I102T exhibited a similar tendency in activity as with the PET substrate, exhibiting the decreased hydrolysis activity. These results indicate that the I102T mutation has a negative effect on catalysis not only on the PET substrate but also on the monomer, BHET.

We present the activity measurements result of both WT and I102T on BHET in this response.

1-5. Compared with CaPETaseWT, the CaPETaseM9 does show great advantages in hydrolytic activity and thermostability. However, the author should highlight the advantages of this rational engineering mutant in the discussion and compare it with other enzyme mutants.

[Response] We described the importance of the final variant in the discussion section (Line #447).

1-6. The discussion is too brief, and the advantages of CaPETase over IsPETase, Tfcutinase, and LCC are not discussed in depth, nor are the reasons for this advantage analyzed in terms of which structural differences give rise to this advantage. What are the implications for the screening and modification of other related enzymes?

[Response] We described the importance of the final variant in the discussion section (Line #418).

Minor concerns

1-1. In this manuscript, there are many formatting errors. For example, “4000g” to “4000 ×g” (line 437), “13500g” to “13500 ×g” (line 441), “mg/ml” to “mg/mL” (line 465), “ml” to “mL” (line 513 and line 514).

[Response] We have revised all of them.

1-2. Please carefully check the references. Such as “line 590”, “line 620”, and “line 663”.

[Response] We have corrected the references. Thank you.

1-3. In supplementary Fig. 2c, SHM40309.1 had an extremely high PET hydrolytic activity, the label 1193.12 coincides with the vertical axis

[Response] In the revised version of the manuscript, we removed the supplementary Fig. 2c and presented in Figure 1b.

1-4. In supplementary Fig. 4, Please carefully check the listed residues, some of them don't seem to match the sequence.

[Response] We have checked the residues and matched it accurately.

1-5. In supplementary Table. 2, please provide CC1/2 values, outer shell values and clash scores in “Refinement”. As well, footnotes are also required for parameters such as R/Rfree values etc.

[Response] We have added the suggested values in the Data collection and Refinement Statistics Table, and the equations defining various R-values are defined in footnotes.

1-6. In line 214, change “PyMol” to “PyMOL”.

[Response] It has been corrected.

1-7. In line 194, change “Fig. 3d, e” to “Fig. 2d, e”.

[Response] It has been corrected.

Reviewer #2 (Remarks to the Author):

The work of Hong et al. provides new results in the field of enzymatic degradation of PET. In this work, a new enzyme from the bacterium *Cryptosporangium aurantiacum* (CaPETase) capable of degrading PET in a wide range of temperatures is characterized, being more active at moderate temperatures than PETase from *Ideonella sakaiensis*, the most active mesophilic enzyme so far. In addition, crystallographic structures of the enzyme are provided showing that CaPETase shows structural differences with other PETases. Furthermore, by combination of point mutations the activity of CaPETase is improved by more than 31-fold at 60°C and the applicability of this new mutant enzyme, CaPETaseM9, in a bioreactor with post-consumer PET is demonstrated. The experimental data is convincing and manuscript is well written and prepared. The data presented in this comprehensive work enriches the scientific field.

We appreciate the reviewer's comments. Thank you for recognizing the implications of our work.

However, to improve the quality of an article, I suggest considering the following:

2-1. In the introduction several PET hydrolyzing enzymes are listed, however there is no mention of the enzyme PHL7, which is the most active thermophilic enzyme so far.

[Response]

We have added PHL7 enzyme to the list of PET hydrolases, citation number #34. (ChemSusChem, 15(9), e202101062) (Line #64)

2-2. The rationale behind the authors' choice of those ten sequences as PETase candidates is unclear.

[Response]

We selected sequences that satisfied the parameters specified in the method section and removed any duplicated or partial genes among them. Then, we randomly selected among the remaining sequences considering the identity values.

Fortunately, we were able to discover enzymes with good PET degradation activity in ambient temperature, and even found new enzymes that show PET degradation activity, although not as good as CaPETase. These results suggested that there is a high possibility of the existence of undiscovered PET-degrading enzymes in nature, and mining for PET hydrolases in biological databases could be a meaningful approach.

2-3. The initial screening of the 10 chosen enzymes solely tests PET hydrolytic activity at 30 °C. If there are any thermophilic enzymes, as in the case of WP 068752972.1, it is not surprising that their activity is very low at 30 °C. It would be interesting to test this enzyme at higher temperatures, closer to its T_m.

[Response]

We thank the reviewer's valuable comments. We performed PET hydrolytic activity measurements for eight enzymes (PC2-PC8, PC10) at various temperatures ranging from 30°C to 60°C, and for a several PET samples (post-consumer PET flake powder, amorphous PET film from Goodfellow, semicrystalline PET powder from Goodfellow), allowing us to obtain more diverse information on each enzyme screened. We have added these results in Fig. 1c.

2-4. It should be noted that the enzymatic activity comparison between CaPETase, LCC, TfCut2 and IsPETase was conducted at a pH of 9.0 with glycine-NaOH buffer, which is not the optimum pH and buffer system for LCC or TfCut2.

[Response]

We agree that performing PET degradation experiments at the optimal conditions for each enzyme

would be the most accurate way to compare their activities. However, it is also noted that Glycine-NaOH pH 9.0 buffer is not the optimal buffer determined by buffer screening for *CaPETase*.

We measured the activity of various enzymes for PET hydrolysis using Glycine-NaOH pH 9.0 buffer, which is commonly used in various previous studies that cover the range of optimal activity for enzymes typically used for PET hydrolysis assay. Furthermore, in the literature reference (*Nat Catal*, 4, 425–430 (2021)/ Refer to Supplementary Fig.8), there were no significant differences in the activity of LCC WT and *TfCut2* WT in Glycine-NaOH pH 8.0 and pH 9.0 buffer conditions, and they used Glycine-NaOH pH 9.0 buffer as the optimal buffer. For these reasons, we considered this comparison to be reasonable.

2-5. There are no such figures referred to as “Fig. 3d” “Fig.3e” and “Fig.3f”.

[Response]

We have corrected the references to Fig.2 instead of Fig.3.

2-6. In line 254, the authors said they introduced 6 point-mutations, i.e. DS2, DS4, NC1, NC4, HP1 and SC2. However, as DS2 and DS4 are double mutants, it would be more accurate to say that eight point-mutations were introduced or 6 mutants were created.

[Response]

We fully agree with the reviewer’s comment. We amended the word to the suggested one “eight”.

2-7. Initial hydrolysis rate values are not provided, but enzyme activity is calculated as the number of products released over a specified period of time. It would be more suitable to provide the values of the initial hydrolysis rates for both the wild-type enzyme and the M9 mutant in order to compare these results with those of other similar enzymes.

[Response]

We have added the initial hydrolysis rates for both the *CaPETase*^{WT} and the *CaPETase*^{M9} at various temperatures, in the revised manuscript. (Supplementary Fig. 19)

Reviewer #3 (Remarks to the Author):

This manuscript describes a novel PET hydrolase and its engineered mutants. The authors determined the crystal structures and analyzed the PET hydrolytic activities in comparison with those of related enzymes. I have some concerns about this manuscript as described below.

Thank you for summarizing our work. We appreciate the reviewer's valuable comments.

3-1. The results of enzyme activity were shown in product amount, the PET monomer concentration. The product concentration would largely depend on the amount applied to HPLC. The yield estimated from the substrate, PET, should be described.

[Response]

As the reviewer suggested, there is a way to calculate the yield by weighing the remaining PET substrate after the reaction. However, we assumed that there would be difficulties in accurately measuring the weight of the remaining substrate in our experimental system. Our experiments mainly used trace amounts of PET substrate in a powdered form (in mg units) not in a film form, so handling the substrate after separation from the reaction mixture and drying posed a risk of experimental error and handling difficulties. Therefore, we presented the enzyme activity based on product chemical quantification through accurate calibration in HPLC.

3-2. The PET degradation would largely depend on the glass transition of PET, as reported previously (Nature 580, 216-219, 2020). The authors should describe the temperature dependent activity in correlation with the glass transition temperature.

[Response]

We agree with the reviewer that the temperature-dependent activity of enzymes related to the glass transition temperature is important in PET degradation, and this phenomenon already described in the Introduction section, line #66.

“For feasible application of enzymatic PET hydrolysis, the inherently high catalytic activity of PET hydrolases is crucial factor as a starting point. Also, exploiting the thermophilic PET hydrolase properties is also an efficient strategy for development of superior PET hydrolases, because high-temperature operation near the glass transition temperature of PET material is favorable to the PET degradation performance.³⁵”

3-3. The section “Discussion” seems to be poor. The authors should revise the section, partly rearranged from the section “Results”.

[Response]

We described the importance of the final variant in the discussion section.

Reviewer #4 (Remarks to the Author):

The paper from Kim and colleagues describes discovery and engineering of a new PET hydrolase. The protein engineering here is good and certainly merits publication in a good journal, but there are many problems with the experimental setup, such that the authors will need to do a major overhaul of this work for this study to be publishable in Nature Communications, mostly related to the inability to replicate the results obtained here and the near-complete lack of substrate characterization. There is little-to-no benchmarking here either to existing enzymes (LCC-ICCG would be sufficient in my opinion). The addition of substrate docking and MD simulations would add a lot of value as well in terms of the (rather over-definitive) interpretation of the differences of CaPETase to existing enzymes.

Thank you for your valuable and insightful comments on our manuscripts. We have conducted additional experiments to address the points you raised and have made overall revisions to the manuscript accordingly.

4-1. The authors should remove the word “novel” from the title and throughout. By definition, original research should be new and novel. See the editorial from S. Scott and C. Jones in ACS Catalysis for a take on this.

[Response]

We have removed the word “novel” from the title and throughout the manuscript.

4-2. Line 23-24 – the authors seem to suggest that engineering has only been done with LCC and IsPETase. I would remove this or substantially change this. Other enzymes have been and are being engineered. These are just the two most prominent to date.

[Response]

We fully agree to the comments. It is true that there has been done a significant amount of enzyme engineering research on many other PET hydrolases as well. In fact, we intended to highlight that a lot of many cases focused on IsPETase and LCC which have superior ability to degradation of PET at mesophilic and thermophilic condition, respectively, in recent research trends. However, I recognized that this statement could be misconstrued. Therefore, I have modified the sentence in lines #23 to accurately convey the intended meaning.

“However, **template enzymes employed in enzyme engineering mainly focused on IsPETase and leaf-branch compost cutinase**, which exhibit extremely mesophilic and thermophilic hydrolytic properties, respectively.”

4-3. Line 27-29 – I think that this is a very “hand-wavy” sentence that does not really convey much useful information.

[Response]

Based on the reviewer’s comments, we revised the sentence “The crystal structure of CaPETase revealed that the high PET hydrolytic activity of the enzyme results from its unique structural features, which are different from other PET hydrolases.” to “We uncovered the crystal structure of CaPETase, which displays a distinct backbone conformation at the active site and residues forming the substrate binding cleft, compared with other PET hydrolases.”.(Line #27)

4-4. Line 49-52 – the authors should either tone down or remove the qualitative and unfounded statements of “sustainability and ecofriendliness”. I would encourage the authors to see Singh et al. Joule 2021 and Uekert et al. Green Chem 2022 on these topics.

[Response]

We thank the reviewer for these valuable comments. We have revised the sentence for refrain the use

of these words.

4-5. Line 62-66 – the authors may want to also see Erikson et al. Nature Comm 2022 and compare their enzyme to the new ones reported therein.

[Response]

We have added the contents related to that reference in the lines #65.

4-6. Line 76-77 – see comment above.

[Response] I have modified the sentence in lines #23 to accurately convey the intended meaning.

4-7. Line 100-102 – why not use a standard and commercially available PET substrate for replication of your results? See ref. 8 for a discussion as to why this is essential. Were these experiments benchmarked to any known PET hydrolases on these inaccessible substrates? I would strongly suggest the authors to use commercial substrates that anyone in the world can purchase and that are well characterized and benchmark to known enzymes on those substrates.

[Response]

We thank the reviewer for the valuable comments.

We have been conducting research on PET-degrading enzymes for a long time, and in the process, we have obtained various PET samples and gained an understanding of the characteristics of various PET samples. The samples we used were PET powder from recycled PET flake, which are actually used as recycled materials for PET. Since there are differences in enzyme activity depending on the type and morphology of the substrate, we believed that the selection of the substrate according to the research purpose was crucial. We have selected PET flake powder, which is a practical recycling material, as the substrate considering its industrial application. And we opted for powdered PET flake (PC-PET) as the substrate, as its uniform morphology and high surface area make it more advantageous for enzymatic reactions, enabling a higher resolution measurement of enzyme activity.

However, we deeply agree with the reviewer's suggestion that we should provide information on substrates that can be commercially used in terms of reproducibility. Therefore, we performed additional experiments using commercially available substrates commonly used in PET degradation experiments with enzymes: Amorphous PET film cat. No. ES301445 (AF-PET, Goodfellow Cambridge Ltd) and Semi-crystalline PET powder cat. No. ES306000 (Cry-PET, Goodfellow Cambridge Ltd).

We have performed additional experiments using commercial PET substrates for comparison of improved enzyme M9 with benchmark enzyme LCC^{ICCG}, comparison with known enzymes, and PC screenings experiments.

4-8. Importantly, there is very little mention of substrate characterization here.

[Response]

In the "Method- Preparation of PET samples" section, we have provided additional information and processing details regarding the substrate preparation and the characterization of these substrates was presented in Supplementary Figs. 27, 28 and Supplementary Table. 3.

4-9. Did the authors do this initial screening of 9 new PETases over T and pH? That should be done. See Erickson et al. 2022. This would add a lot of value beyond just CaPETase.

[Response]

We appreciated the reviewer's valuable comments. I agree that the evaluation of the PET hydrolytic activity under various conditions of the new enzymes can provide valuable information. We additionally have conducted a screening experiment using eight PETase candidates on various PET substrates (PC-PET, AF-PET, Cry-PET), temperature ranging from 30°C to 60°C, and various pH

(pH6.0 to pH10.0). These results of the experiment were presented in the Fig.1c and Supplementary Fig. 3.

4-10. For the sequence homology searches, there are published tools to predict the optimal temperature of an enzyme – did the authors look into this at all?

[Response]

In fact, we didn't investigate it beforehand.

When we discovered new enzymes, we placed importance not only on their thermostability, but also on the inherent activity of the enzyme itself (activity at the ambient temperature condition). Therefore, since our objective was not solely to discover thermophilic enzymes, we did not perform investigation on the optimal temperature of the enzyme before performing further experiments.

However, we thought that such a tool (TOME) could be very helpful in efficient screening for enzymes in the future. (*Journal of Chemical Information and Modeling*, 2020, 60(8), 4098-4107) Thank you for the valuable information.

4-11. Line 120-121 – it is not surprising at all that TfCut2 and LCC have low activity at 30°C. This is already known.

[Response]

We revised the sentence. (Line #131)

4-12. Figure 1 – please put the reaction conditions (enzyme mass loading per substrate mass loading, solids loading, buffer, substrate characteristics, etc.) in the figure caption.

[Response]

We added the contents of reaction conditions to the figure legend.

4-13. Line 156-157 – Since the authors make such a big deal of this difference, why put the images that show this in the SI?

[Response]

We thought that the structural differences of CaPETase we proposed are adequately demonstrated in Fig. 2a (revised manuscript version), where the torsion angle differences are depicted as a putty tube representation. In addition, supplementary Fig. 7 provided more clear illustration of the structural differences of mainchain at the backbone level.

4-14. The structural description for the backbone dihedral angles presupposes that the apo-structure solved here is indicative of the PET binding conformation. Were any MD simulations conducted to see the flexibility of the places in the structure where these differences are noted? Did the authors do any intrinsic flexible docking simulations of a PET oligomer?

[Response]

Thank you for the valuable comments. We have conducted molecular dynamics simulations for CaPETase, IsPETase, TfCut2, and LCC to obtain backbone fluctuation profiles. Based on our MD results, we identified regions in which CaPETase differs in flexibility and stability from other enzymes, which supports the previously described differences in backbone torsion angle at the active site of CaPETase. We have added these findings in the revised manuscript.

4-15. The structural paragraph repeatedly refers to Figure 3, but I think the authors mean Figure 2.

[Response]

We have corrected it.

4-16. Figure 2b does not convey much information that is useful.

[Response]

We have replaced the Fig. 2b to Supplementary Fig. 8.

4-17. Figure 2 and the surrounding structural analysis would be greatly strengthened by the addition of intrinsic flexible docking of a PET oligomer into the active site and comparison of that with previously published docking studies.

[Response]

We have attempted docking simulation for the PET substrate and were able to successfully perform docking simulations for the MHET monomer, building block of PET polymer. However, we encountered difficulties in obtaining accurate docking results for the PET oligomers, which is actual substrate of PET hydrolase.

4-18. Reaction conditions should be listed in the Figure 2 caption.

[Response]

We added the contents of reaction conditions to the figure legend.

4-19. Line 232 – same comment as above – I VERY strongly urge the authors to reconsider their substrate choice and to redo these experiments with substrates that are commercially available and well characterized as a benchmark for these types of experiments.

[Response]

We thank the Reviewer for pointing out this issue. As mentioned above, we have performed additional experiments using commercial PET substrates purchased from Goodfellow Cambridge Ltd for comparison of CaPETase^{WT} and CaPETase^{M9}, comparison of improved enzyme M9 with benchmark enzyme LCC^{ICCG}, comparison with known enzymes, and PC screenings experiments. We have added these experimental data to the revised manuscript. However, we did not perform additional experiments on variants designed through rational engineering, as they were designed to verify the effects of PET degradation and thermal stability, and we thought that they would show similar trends even with different type PET substrates.

4-20. Line 295-296 – same comments as before – this is nice protein engineering work, but it is not conducted on a substrate that is available for the community, and thus, this work is not reproducible.

[Response]

As mentioned above, we have performed additional experiments using commercial PET substrates (Cry-PET) purchased from Goodfellow Cambridge Ltd for comparison of benchmark enzyme, LCC^{ICCG} and CaPETase^{M9}. We have added these experimental data to the Figure. 3c.

4-21. Figure 3 – Same comments as above; please include the reaction conditions in the caption.

[Response]

We have added the contents of reaction condition to the Figure legend.

4-22. Overall, it would be much better if the authors would report % conversion instead of product amount in all figures. Are the authors only including TPA in the product release? Does this include MHET and BHET? Are those being measured and if so, what is the selectivity to TPA versus BHET and MHET?

[Response]

When considering the conditions for the most of our PET degradation experiments conducted in e-tube vials, excluding reactor systems or dialysis systems, we thought that achieving complete degradation is a limit due to factors such as concentration of the buffer solution, the pH decrease caused by decomposition products, and the solubility issues of reaction products at different temperatures.

Therefore, instead of focusing on the conversion rate, the concentration of decomposition products was presented as an indication of the capacity of the enzyme to show its activity.

The concentration of the decomposition products was presented as the sum of TPA, MHET, and BHET released during the reaction. While the ratio of these products varies depending on the reaction temperature and time, in most cases, BHET is either very low or absent, and TPA and MHET are present in a ratio of approximately 1:1 to 2:1.

4-23. How did the authors choose the times they are running the reactions as well? Does the product release plateau or cease at the chosen reaction time(s)? How does the pH change in these screening reactions not done in a pH-controlled system?

[Response]

The time course reactions were monitored until the point where the rate of product release decreased significantly or reached a plateau. In the reaction for screening of the variants, the reaction time was carefully chosen based on the buffer capacity and the time point where the initial rate of the enzyme reaction could be appropriately determined at each temperature.

We have perceived that the pH changes occur during the reaction in a system without pH control. Therefore, the PET degradation experiment was performed within a range where the pH does not change significantly. In fact, we confirmed that under the conditions of 50 mM Glycine-NaOH pH 9.0, the buffering capacity prevented significant changes in pH up to a concentration of 5 mM of reaction products.

4-24. Line 341-343 – same comments as above – this work is not reproducible. Please repeat this work with a substrate from e.g. Goodfellow or another commercial vendor and thoroughly characterize the substrate.

[Response]

As answered above, we have performed additional experiments using commercial PET substrates purchased from Goodfellow Cambridge Ltd for comparison of CaPETase^{WT} and CaPETase^{M9}, comparison of improved enzyme M9 with benchmark enzyme LCC^{ICCG}, comparison with known enzymes, and PC screenings experiments. Our results confirmed that the trends in enzymatic activity for PC-PET and Cry-PET remained consistent. Therefore, we presented the experiment's results in the reactor using PC-PET, taking into account the industrial application of PET degradation. Additionally, we also included information on the characteristics of PC-PET and its preparation in the Method section (Line #462) and Supplementary Figs. 27, 28 and Supplementary Table. 3.

4-25. Did the authors do a thorough pH/buffer/T screen of the best enzyme and the wild-type enzyme to find the optimal conditions for each? This is not clear to me.

[Response]

We conducted screening experiments with various pH buffers and temperatures for CaPETase^{WT} and CaPETase^{M9}, and included them in the supplementary Figs. 3 and 19.

4-26. Line 350-352 – I do not know what the authors are trying to convey here. Also, comparing their results to those from Tournier et al. are challenging vis-a-vis substrate inconsistencies.

[Response]

Since the properties of PC-PET substrate used in our reactor system are similar to those from Tournier et al (Crystallinity: ~10%; Particle-size distribution: D90 < 500 μm, 200 μm < D50 < 250 μm), we thought that comparing the performance of PET degradation in similar reactor systems is a reasonable comparison. We would like to emphasize that the PET degradation ability of CaPETase^{M9} performed at a relatively lower temperature condition of 55°C in a bioreactor showed comparable performance to the PET degradation performed at a favorable T_g temperature, considering the influence of temperature-

dependent activity in the enzymatic PET degradation. However, we agree with the reviewer's opinion that it is challenging to make direct comparisons under equivalent conditions because the samples are similar but not identical. We have revised this sentence.

4-27. Line 355 – why is this known? Citation?

[Response]

Colored PET is generally considered to be disadvantageous for mechanical and chemical recycling techniques, with the exception of enzymatic recycling. This is because non-transparent colored bottles and non-transparent white bottles have limited market value, as they may contain pigments that cause contamination during recycling or may result in undesirable colors when mixed with clear or colored PET. The relative reference has been cited #45.

4-28. Line 359 – are you sure that the differences (which are slight) are because of the impurities? Did you control for particle size, PET MW, crystallinity, etc.? Those factors would have to be exactly controlled for to make this claim.

[Response]

The processing of PC-PET^{Transparent} and PC-PET^{Colored} substrates was carried out identically, and the particle size of the substrates was also controlled to be under 200 μ m. Additionally, we confirmed that the properties of the two materials as a substrate were similar through crystallinity measurements, with PC-PET^{Transparent} at 9.9% and PC-PET^{Colored} at 10%. Therefore, we believed that such a phenomenon was caused by factors other than the conditions suggested by the reviewer. However, as this claim has not been proven and is based on speculation, we have modified the sentence. (line #394)

4-29. Line 362 – again, what substrate is this? Is it available for anyone? Is it well characterized?

[Response]

This substrate is a commercially available salad container, and the experiment was conducted using untreated post-consumer PET to demonstrate the applicability of PET enzymes. The results showed significant degradation activity even on untreated post-consumer PET, highlighting its potential for application. Various post-consumer plastics were experimented on to emphasize the importance of this aspect. The crystallinity of this substrate is presented in Supplementary Fig. 28.

4-30. Line 375-376 – please remove claims of eco-friendly and energy-efficient – see analysis studies cited above. This process actually has a long way to go yet to truly be “energy-efficient” and “ecofriendly”.

[Response]

We have removed the word “energy-efficient” and “ecofriendly” and edited the sentence. (line #408)

4-31. Methods – Preparation of PET samples – this is not sufficient at all. What is the crystallinity? Mw and Mn? Particle size distribution? See ref. 8. The authors really ought to use Goodfellow or other chemical vendor-supplied PET. Calling their substrate B-PET is a misnomer. It originated from bottles, yes, but after cryo milling, it will not be bottle-grade PET anymore.

[Response]

We thank the reviewer for the valuable comments. To provide more detailed information about the PET samples used in this study (PC- PC-PET^{Transparent} and PC-PET^{Colored}), we conducted additional analyses including particle size distribution and GPC (Gel Permeation Chromatography). We have provided additional information on the particle size distribution analysis in Method section and have also elaborated on the preparation of the PET samples. Additionally, the crystallinity of each PET sample is presented in Supplementary Fig. 28, and the Mw and Mn are included in Supplementary Table. 3.

We conducted additional experiments using PET samples (Product No. ES301445 and ES306000) purchased from Goodfellow in response to the comment suggesting the use of commercially available PET samples. The results of these experiments are presented throughout the revised manuscript. And we fully agree with the third comments. In the revised manuscript, however, only the results obtained for PC-PET, AF-PET, and Cry-PET were presented to ensure consistency of the PET samples during the re-experiments, except for the B-PET. Therefore, the term "B-PET" was not used in this manuscript.

REVIEWER COMMENTS

Reviewer #1 (Remarks to the Author):

The authors have taken into account the suggestions provided in the review and have made significant improvements to several aspects of the manuscript. They have clarified the rationale behind introducing additional disulfide bonds and summarized the effects of different positions of disulfide bonds on activity and thermal stability. Furthermore, the authors have conducted experiments to test the activity of CaPETase at different temperatures and with different ions, as suggested by the reviewers.

In response to the comment about the unique structural characteristics of CaPETase, the authors have provided a thorough analysis of the backbone torsion angles at the connecting loops, highlighting the differences between CaPETase and other PET hydrolases. They have also explored the substrate selectivity of CaPETase through the use of additional substrates and discussed the implications of their findings.

I think there are no major problems in the whole paper, but some minor errors are still need to be carefully checked before publication.

1. MD procedure details such as software or online sites for protein protonized, protein energy minimization should be added.
2. In Fig1, "Discovery a PET hydrolase and initial characterizations of CaPETase." should be "The discovery of xxx".
3. In line 468-470, there should be a space between numbers and units, please check full manuscript carefully.
4. In line 572, "Na⁺ and Cl⁻" should superscript.
5. In manuscripts, some use "15 mg mL⁻¹", while others use 15 mg/mL, Please unify the unit format.
6. Pay attention to spelling such as In line 443: "noteworty" to "noteworthy".

Reviewer #2 (Remarks to the Author):

1. It is recommended that the authors provide the rationale behind the selection of the ten sequences in the methodology section.
2. The standard deviation of the activity of PC2 with AF-PET at 40 °C in Fig. 1c appears to be excessively high. This substantial variability is of particular significance considering the authors' emphasis on the high activity of CaPETase at moderate temperatures.
3. The initial rates have been expressed as uM/h by dividing the uM released by the duration of the experiment. However, it is recommended to measure additional time points. For instance, previous observations LCC showed an increase in the specific activity between 8 and 16 hours of reaction (Sonnendecker et al. 2021). Additionally, it would be beneficial to express the enzymatic activity as specific rates, such as umol/h/mg enzyme or a similar unit, which would account for the amount of enzyme present and provide a more meaningful measure of enzymatic efficiency.
4. In line 134, where it states, "In particular, the PET hydrolytic activity of CaPETase was 11.5-fold higher than that of IsPETase (Fig. 1c)," it is not mentioned at which temperature this activity was observed. It is presumed to be 40 °C.
5. There are errors in figure references as in the previous version. In line 132, 134, 135 it says Fig. 1c while it should refer to Fig. 1d instead. I kindly request the authors to exercise greater care in accurately referencing the figures throughout the manuscript to facilitate the revision process.
6. There is a discrepancy between the previous version, where the activity at 40 °C was reported as 7.5-fold higher than that of IsPETase, and the current version, which states an 11.5-fold increase. However, in Figure 1d, the activity of CaPETase at 40 °C does not exhibit an 11.5-fold increase compared to IsPETase. It is important for the authors to provide clarification regarding these

discrepancies between the reported numbers and the data presented in Figure 1d.

Reviewer #3 (Remarks to the Author):

The authors revised the manuscript well.

Reviewer #4 (Remarks to the Author):

The authors have fully addressed my comments. I think that this paper is acceptable for Nature Communications now!

Authors' point-by-point responses

Reviewer #1 (Remarks to the Author):

The authors have taken into account the suggestions provided in the review and have made significant improvements to several aspects of the manuscript. They have clarified the rationale behind introducing additional disulfide bonds and summarized the effects of different positions of disulfide bonds on activity and thermal stability. Furthermore, the authors have conducted experiments to test the activity of CaPETase at different temperatures and with different ions, as suggested by the reviewers.

In response to the comment about the unique structural characteristics of CaPETase, the authors have provided a thorough analysis of the backbone torsion angles at the connecting loops, highlighting the differences between CaPETase and other PET hydrolases. They have also explored the substrate selectivity of CaPETase through the use of additional substrates and discussed the implications of their findings.

I think there are no major problems in the whole paper, but some minor errors are still need to be carefully checked before publication.

[Response] We appreciated the reviewer for taking time carefully review the revised manuscript. Below is our point-by-point response to each comment.

1.MD procedure details such as software or online sites for protein protonized, protein energy minimization should be added.

[Response] We added the information on software or online site used for protein protonation (propka3), protein energy minimization (gromacs with conjugate gradient algorithm) to the methods section. **(Line #571-574)**

2.In Fig1,"Discovery a PET hydrolase and initial characterizations of CaPETase."should be "The discovery of xxx".

[Response] We have revised the “Discovery” to “The discovery ~”. **(Line #145)**

3.In line 468-470, there should be a space between numbers and units, please check full manuscript carefully.

[Response] In revised manuscripts, it has been carefully checked and corrected.

4.In line 572, “Na⁺ and Cl⁻” should superscript.

[Response] It has been corrected. **(Line #573)**

5. In manuscripts, some use “15 mg mL⁻¹”, while others use 15 mg/mL, Please unify the unit format.

[Response] It has been corrected.

6. Pay attention to spelling such as In line 443: “noteworty” to “noteworthy”.

[Response] It has been corrected.

Reviewer #2 (Remarks to the Author):

1. It is recommended that the authors provide the rationale behind the selection of the ten sequences in the methodology section.

[Response] We have described the rationale for selecting the sequences in the Methods section - Selection of PETase candidates and phylogenetic analysis. **(Line #479-484)**

2. The standard deviation of the activity of PC2 with AF-PET at 40 °C in Fig. 1c appears to be excessively high. This substantial variability is of particular significance considering the authors' emphasis on the high activity of CaPETase at moderate temperatures.

[Response] Thanks for pointing out this. To ensure the reliability of this results, we have performed re-measurement of the PET hydrolytic activity of PC2 with AF-PET at 40 °C. Three independent experiments were conducted, and the obtained data represented a decrease in the standard deviation. This reduction in standard deviation signifies an improvement in the accuracy and consistency of this data. This result presented in Fig. 1c of the revised manuscript.

3. The initial rates have been expressed as uM/h by dividing the uM released by the duration of the experiment. However, it is recommended to measure **additional time points**. For instance, previous observations LCC showed an increase in the specific activity between 8 and 16 hours of reaction (Sonnendecker et al. 2021). Additionally, it would be beneficial to express the enzymatic activity as specific rates, such as **umol/h/mg enzyme** or a similar unit, which would account for the amount of enzyme present and provide a more meaningful measure of enzymatic efficiency.

[Response] We appreciated the reviewer's valuable comments. As the reviewer suggested, we have modified the enzymatic activity unit of CaPETase WT and CaPETase M9 at 30-70 °C from “initial hydrolysis rate (uM/h)” to “specific rate (umol/h/mg_{enzyme})”. (Supplementary Fig.20). Additionally, we have conducted further experiments to profile the specific activity of CaPETase WT and CaPETase M9 at different reaction time point. These experiments were performed at 40 °C and 60 °C, which are the optimal temperatures for each enzyme. These results of the experiment were presented in the Supplementary Fig. 20.

4. In line 134, where it states, "In particular, the PET hydrolytic activity of CaPETase was 11.5-fold higher than that of IsPETase (Fig. 1c)," it is not mentioned at which temperature this

activity was observed. It is presumed to be 40 °C.

[Response] Thank you for pointing out this. We have corrected the description as “In particular, the PET hydrolytic activity of *Ca*PETase was 3.1-fold higher than that of *Is*PETase at 40 °C” in the revised manuscript. **(Line #135)**

5. There are errors in figure references as in the previous version. In line 132, 134, 135 it says Fig. 1c while it should refer to Fig. 1d instead. I kindly request the authors to exercise greater care in accurately referencing the figures throughout the manuscript to facilitate the revision process.

[Response] We thank the reviewer for pointing out our mistake. It has been corrected, and figure references have been checked accurately throughout the manuscript.

6. There is a discrepancy between the previous version, where the activity at 40 °C was reported as 7.5-fold higher than that of *Is*PETase, and the current version, which states an 11.5-fold increase. However, in Figure 1d, the activity of *Ca*PETase at 40 °C does not exhibit an 11.5-fold increase compared to *Is*PETase. It is important for the authors to provide clarification regarding these discrepancies between the reported numbers and the data presented in Figure 1d.

[Response] We sincerely apologize for the mistake, and we have adjusted the values in our revised manuscript. Regarding the observed discrepancies in certain numerical values compared to the previous version, we would like to clarify that during the first revision process, we conducted additional experiments using commercial PET substrates purchased from Goodfellow Cambridge Ltd for comparison of PET hydrolytic activity of *Ca*PETase, *Is*PETase, LCC, and *Tj*Cut2 as requested by reviewers. The updated results from the new experiments have been included in Figure 1d, replacing the previous version. The change in PET substrate source have introduced variations in the measured values, leading to differences between the previous and current results.

Reviewer #3 (Remarks to the Author):

The authors revised the manuscript well.

Reviewer #4 (Remarks to the Author):

The authors have fully addressed my comments. I think that this paper is acceptable for Nature Communications now!

REVIEWERS' COMMENTS

Reviewer #2 (Remarks to the Author):

The authors have responded satisfactorily to my remarks.